# SE-Agent: Self-Evolution Trajectory Optimization in Multi-Step Reasoning with LLM-Based Agents

**Yifu Guo**[1,2*]    **Jiaye Lin**[3*]    **Huacan Wang**[4*†]    **Yuzhen Han**[5]
**Sen Hu**[6]    **Ziyi Ni**[4,7]    **Licheng Wang**[4]    **Mingguang Chen**[8]

[1]Sun Yat-sen University, [2]StepFun, [3]Tsinghua University,
[4]University of Chinese Academy of Sciences, [5]University of Toronto, [6]Peking University,
[7]Institute of Automation, Chinese Academy of Sciences, [8]University of California, Riverside

✉ quantaalpha.ai@gmail.com    ⌦ JARVIS-Xs/SE-Agent

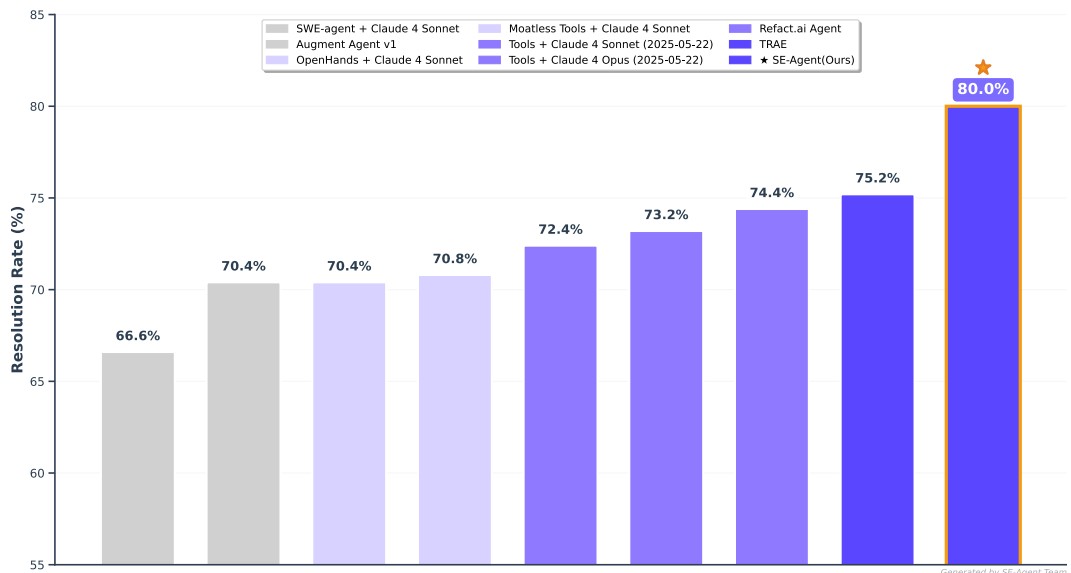

**Code Agent Resolution Rate Comparison (SWE-Bench Verified)**

## Abstract

Large Language Model (LLM)-based agents have recently shown impressive capabilities in complex reasoning and tool use via multi-step interactions with their environments. While these agents have the potential to tackle complicated tasks, their problem-solving process—agents' interaction trajectory leading to task completion—remains underexploited. These trajectories contain rich feedback that can navigate agents toward the right directions for solving problems correctly. Although prevailing approaches, such as Monte Carlo Tree Search (MCTS), can effectively balance exploration and exploitation, they ignore the interdependence among various trajectories and lack the diversity of search spaces, which leads to redundant reasoning and suboptimal outcomes. To address these challenges, we propose SE-Agent, a **S**elf-**E**volution framework that enables **Agents** to optimize their reasoning processes iteratively. Our approach revisits and enhances former pilot trajectories through three key operations: revision, recombination, and refinement. This evolutionary mechanism enables two critical advantages: (1) it expands the search space beyond local optima by intelligently

---

*Equal Contribution.
†Corresponding Author.

39th Conference on Neural Information Processing Systems (NeurIPS 2025).

exploring diverse solution paths guided by previous trajectories, and (2) it leverages cross-trajectory inspiration to efficiently enhance performance while mitigating the impact of suboptimal reasoning paths. Through these mechanisms, SE-Agent achieves continuous self-evolution that incrementally improves reasoning quality. We evaluate SE-Agent on SWE-bench Verified to resolve real-world GitHub issues. Experimental results across five strong LLMs show that integrating SE-Agent delivers up to **55%** relative improvement, achieving **state-of-the-art** performance among all open-source agents on SWE-bench Verified (61.2% with Claude-3.7-Sonnet, 80.0% with Claude-4-Sonnet[1]).

## 1 Introduction

Large Language Models (LLMs) have demonstrated remarkable capabilities in various domains, from natural language understanding to code generation [1]. When equipped with external tools [2; 3; 4; 5] and environmental interaction capabilities, these models evolve into autonomous agents that can tackle increasingly complex real-world tasks.

However, completing complex tasks rarely happens in a single step [6; 7]. In practice, most LLM-based agents employ multi-turn interactions with their environments, following frameworks like ReAct [8] that iteratively gather information, reason about the current state, and take actions. These interaction processes naturally form trajectories—sequences of states and actions that encode valuable problem-solving patterns and strategies [9; 10]. Each trajectory represents a complete attempt at solving a given problem, encompassing not just the final solution but also the reasoning path, environmental feedback, and decision-making process that led to the outcome [11; 12; 13].

Despite the wealth of information contained in these interaction trajectories, current approaches to multi-agent reasoning remain fundamentally limited [14; 15]. While methods such as Monte Carlo Tree Search (MCTS) effectively balance exploration and exploitation [16; 17], they treat trajectories as independent entities, ignoring the rich interdependencies and potential synergies among different solution paths [18]. Moreover, even when employing diverse sampling strategies (e.g., varying temperature parameters or prompts), agents tend to converge on structurally similar trajectories that differ only in surface-level expressions, leading to a critical phenomenon: despite generating multiple trajectories, the final outcomes remain surprisingly homogeneous [19; 20; 21]. This limitation stems from the inherent nature of probabilistic language models, which naturally gravitate toward high-probability solution patterns, thereby constraining the diversity of the search space [22; 23; 24].

To overcome these limitations, we propose SE-Agent, a **S**elf-**E**volution framework that enables **Agents** to iteratively optimize their reasoning processes through systematic trajectory manipulation. Our key insight is that by actively intervening at the trajectory level—rather than merely adjusting sampling parameters—we can guide agents to explore fundamentally different perspectives and solution approaches. Through three core operations (revision, recombination, and refinement), SE-Agent not only generates genuinely diverse trajectories but also produces correspondingly diverse outcomes, significantly expanding the candidate solution space. This trajectory-level intervention enables agents to discover novel problem-solving capabilities that may not emerge from conventional sampling methods, effectively allowing base models to transcend their initial performance boundaries. By strategically combining insights from multiple trajectories, our framework amplifies the likelihood of finding correct solutions to challenging problems that would remain unsolved through traditional multi-sampling approaches. Our contributions are summarized as follows:

- We introduce a novel self-evolution framework that operates at the trajectory level to enhance agent reasoning capabilities. Importantly, our approach remains effective regardless of improvements in base model capabilities, as long as complex tasks continue to require multi-step reasoning—a requirement likely to persist in the foreseeable future. By manipulating trajectories rather than relying on sampling variations, we achieve genuine diversity in solution paths and final outcomes.

- We conduct comprehensive experiments on SWE-bench Verified [25], one of the most challenging and widely-adopted benchmarks for code-related tasks. Our results demonstrate significant performance improvements across different LLMs, validating the effectiveness of trajectory-level self-evolution in real-world software engineering scenarios.

---

[1]80.0% is achieved with Claude-4-Sonnet using the latest SWE-Agent (aligned with the May 22, 2025 SWE-bench Verified leaderboard; baseline 66.6% resolved) together with our SE-Agent.

## 2  Related Works

**Code Agents**  Code agents represent a specialized class of AI systems designed to understand, generate, and manipulate source code autonomously. These agents have evolved to handle increasingly complex software engineering tasks within large-scale codebases. Given repository-level objectives, they identify relevant files and code segments before implementing necessary modifications. In this work, we focus on the SWE-bench task, which involves resolving real-world GitHub issues by automatically applying functional bug fixes. [26] introduces the concept of agent-computer interfaces through SWE-agent, while OpenDevin [27] presents a collection of community-driven agents, including CodeAct [28]. The Agentless approach [29] achieves competitive performance using a streamlined two-step process of localization and repair. AutoCodeRover [30] incorporates advanced code analysis techniques, including abstract syntax trees and spectrum-based fault localization. Alibaba's Lingma Agent [31] proposes a search-based strategy for repository exploration followed by structured editing. Additionally, several studies [32; 33; 34; 35] demonstrate that repeated trajectory sampling, even under identical agent configurations, can lead to significant variance in outcomes. More recently, SWE-search [36] proposes a multi-agent framework integrating Monte Carlo Tree Search (MCTS) with a self-improvement mechanism to enhance performance on such tasks.

**Agent Capability Enhancement**  Recent research has developed diverse approaches to enhance intelligent agent performance. Planning frameworks like GoalAct [37] introduce global planning with hierarchical execution, reducing complexity and improving adaptability by 12.22% on LegalA-gentBench [38]. For code generation, the RGD framework [39] leverages multi-agent debugging for iterative optimization, outperforming state-of-the-art methods by 9.8% on HumanEval and 16.2% on MBPP datasets. Collaborative approaches such as Collaborative Voyager [40] enable agents to communicate and learn from each other, effectively addressing hallucinations while enhancing task completion. Meta-planning optimization through MPO [41] provides high-level guidance and continuously optimizes plans based on execution feedback, significantly improving task efficiency and generalization. Agent augmentation methods like AutoGPT [42] and AgentGPT [43] integrate tool usage to expand agent capabilities, while retrieval-augmented frameworks such as MemGPT [44] and ReAct [8] enhance contextual understanding through memory mechanisms. Self-improvement techniques, including Reflexion [10] and CRITIC [45], enable agents to iteratively refine their reasoning through self-critique. While these methods show promise, our work introduces a novel approach within the ReAct paradigm that incorporates strategic reflection and mutation at critical steps, combining multiple trajectories to generate optimized execution paths without requiring extended computation time like Test-Time Scaling (TTS) techniques.

## 3  Preliminaries and Problem Setup

**Task-Oriented Reasoning Environment**  We consider a general class of complex tasks that require multi-step reasoning and execution. These tasks can range from software engineering problems to mathematical problem-solving, strategic planning, or creative content generation. Formally, we model the reasoning environment as a tuple $\mathcal{E} = (\mathcal{T}, \mathcal{S}, \mathcal{A}, \mathcal{P}, \mathcal{R})$, where $\mathcal{T}$ represents the space of all possible tasks that require multi-step reasoning $\mathcal{S}$ denotes the state space, where each state $s \in \mathcal{S}$ captures the current progress toward solving a task; $\mathcal{A}$ is the action space available to the agent, which may include reasoning steps, information gathering, or direct task execution; $\mathcal{P} : \mathcal{S} \times \mathcal{A} \to \mathcal{S}$ defines the transition dynamics that map a state-action pair to a new state; $\mathcal{R} : \mathcal{S} \times \mathcal{T} \to \mathbb{R}$ is the reward function that evaluates the quality of a state for a given task.

**Reasoning Trajectories**  Central to SE-Agent is the concept of a reasoning trajectory, which represents the sequential progression of states and actions as the agent works toward solving a task. Given a task $t \in \mathcal{T}$, a reasoning trajectory $\tau$ is defined as an ordered sequence: $\tau = (s_0, a_0, s_1, a_1, \ldots, s_n)$ where $s_0$ is the initial state, $a_i$ is the action taken at step $i$, and $s_n$ is the final state. Each intermediate state is determined by the transition function: $s_{i+1} = \mathcal{P}(s_i, a_i)$. The trajectory $\tau$ is generated by repeatedly applying a policy $\pi : \mathcal{S} \times \mathcal{T} \to \mathcal{A}$, which maps the current state and task to an appropriate action. The policy can incorporate various reasoning strategies, including decomposition, planning, and verification. The quality of a trajectory is measured by the final reward: $R(\tau, t) = R(s_n, t)$, which evaluates how well the final state $s_n$ satisfies the requirements of task $t$.

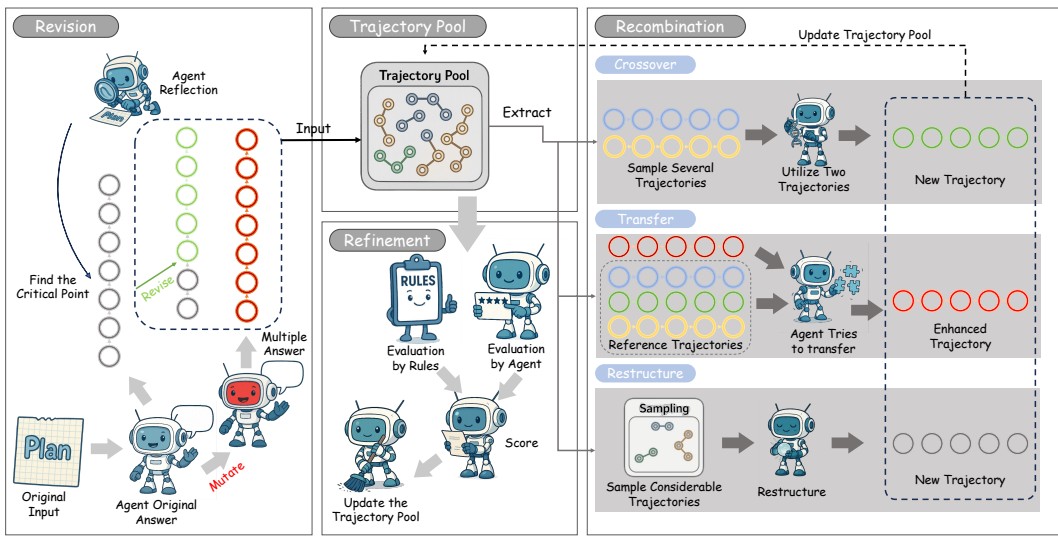

Figure 1: **Overview of the proposed *SE-Agent* self-evolution framework.** Starting from an initial pool of diverse pilot trajectories, the agent iteratively performs three trajectory-level operators—*Revision*, *Recombination*, and *Refinement*—to harvest cross-trajectory insights, escape local optima, and converge to a high-reward solution path that robustly solves the target task.

**Objective of Agent** The objective of our work is to develop an agent that can generate high-quality reasoning trajectories for complex tasks. Specifically, given a task $t \in \mathcal{T}$, we aim to find a policy $\pi^*$ that maximizes the expected reward: $\pi^* = \arg\max_\pi \mathbb{E}_{t \sim \mathcal{T}}[\mathcal{R}(\tau^\pi(t), t)]$, where $\tau^\pi(t)$ denotes the trajectory generated by policy $\pi$ for task $t$.

# 4 SE-Agent

In this section, we introduce SE-Agent, a novel yet self-evolution paradigm to tackle intricate tasks involving multi-step executions. To derive a high-rewarding trajectory, SE-Agent alternatively generates a series of improved trajectories in which the best one is chosen. Specifically, we design the trajectory evolution mechanism using pre-collected pilot trajectories, which generate references for imitation learning of the SE-Agent. Our main inspiration is to formulate the pilot trajectories as an "improvement operator" applied to the apprentice trajectory, which allows the agent to continuously evolve its reasoning paths through iterative refinement and cross-trajectory learning.

## 4.1 Overview of SE-Agent

The core philosophy of SE-Agent lies in leveraging the collective intelligence embedded within multiple reasoning trajectories to transcend the limitations of isolated reasoning attempts. As illustrated in Figure 1, SE-Agent operates through an evolutionary framework that systematically improves trajectory quality across iterations.

Given a task $T$, we first generate a diverse pool of initial trajectories $\mathcal{T}_0 = \{t_1, t_2, ..., t_n\}$ through multi-dimensional planning and exploration. Each trajectory $t_i$ represents a sequence of reasoning steps and actions taken by the agent to solve the task $T$. Rather than selecting the best trajectory from this initial pool and terminating, as traditional approaches do, SE-Agent employs an iterative evolution process to derive increasingly improved solutions.

Our SE-Agent repeats the following three fundamental operations:

- **Revision:** Enhancing individual trajectories through self-reflection and targeted improvement
- **Recombination:** Creating new trajectories by combining strengths from existing paths
- **Refinement:** Optimizing trajectories by eliminating redundancies and enhancing efficiency

Each iteration $i$ produces a new generation of trajectories $\mathcal{T}_i$, with the quality of solutions progressively improved(See Fig 6 for a detailed case study demonstrating this process on a real Astropy bug fix.). This process continues until the convergence criteria are met or a predetermined number of iterations is reached. The final output is the highest-rewarding trajectory $t^*$ that most effectively solves the original task. The key innovation of SE-Agent is its ability to escape local optima by intelligently exploring the solution space guided by previous experiences while simultaneously leveraging cross-trajectory inspiration to efficiently enhance performance. This dual mechanism enables continuous self-evolution of the agent's reasoning capabilities.

One may interpret SE-Agent as a specialized form of genetic algorithm tailored for large language model reasoning. From such a perspective, our approach shares conceptual similarities with evolutionary computation frameworks where trajectory generation acts as the genotype, and the problem-solving performance represents the phenotype expression. However, unlike traditional genetic algorithms that typically require numerous iterations to converge to acceptable solutions, SE-Agent is designed to achieve high-quality results with remarkably fewer evolutionary cycles. This efficiency stems from our targeted operations that leverage the inherent reasoning capabilities of LLMs combined with structured evolution mechanisms. Furthermore, SE-Agent bears resemblance to self-play and expert iteration approaches in reinforcement learning, where each trajectory refinement step serves as an improvement operator that guides subsequent reasoning through enhanced exploration and exploitation balance. The key distinction lies in our explicit manipulation of complete reasoning trajectories rather than isolated state-action pairs, enabling more holistic improvements across the entire problem-solving process.

## 4.2 Revision Operation

The revision operation forms the foundation of SE-Agent's self-evolution capability, focusing on generating and improving individual trajectories through introspection and targeted enhancement.

### 4.2.1 Generating Initial Trajectories

To establish a diverse starting point for evolution, we employ two complementary approaches.

**Multi-Planning Exploration** We first generate several distinct trajectories for task $T$ by varying planning strategies, prompting techniques, and reasoning approaches. This maximizes the dimensional diversity of our initial trajectory pool, ensuring broad coverage of the solution space. Each trajectory $t_i \in \mathcal{T}_0$ is generated as

$$t_i = \text{Plan}(T, \theta_i),$$

where $\theta_i$ represents different planning parameters and strategies.

**Mutation-Based Diversification** We further expand the trajectory pool by applying controlled mutations to existing trajectories, producing $M$ additional paths. These mutations introduce targeted variations in reasoning steps, action selections, or intermediate conclusions:

$$t_{i+m} = \text{Mutate}(t_i, \delta_m), \;\; n = 1, \ldots, M$$

where $\delta_n$ controls the degree and nature of mutations applied.

This dual approach results in an initial pool of ten diverse trajectories $\mathcal{T}_0 = \mathcal{T}_0^{plan} \cup \mathcal{T}_0^{mutate}$ that serve as the foundation for subsequent evolution.

### 4.2.2 Reflection and Revision

For each trajectory $t_i \in \mathcal{T}_0$, we perform a critical reflection process that analyzes strengths, weaknesses, and potential improvement areas:

$$R_i = \text{Reflect}(t_i, T).$$

Based on these reflections, we derive revised trajectories through targeted improvements:

$$t_i' = \text{Revise}(t_i, R_i).$$

The reflection process identifies logical inconsistencies and elaborates on underdeveloped reasoning steps. The revision process then eliminates redundant or circular reasoning and incorporates alternative perspectives when beneficial.

The revision operation embodies the principle of "planning origin and reflective evolution", where initial plans serve as seeds that evolve through structured self-reflection and targeted improvement.

### 4.3 Recombination Operation

While the revision operation enhances individual trajectories, the recombination operation facilitates collective evolution through cross-trajectory learning.

#### 4.3.1 Trajectory Recombination

We implement three complementary recombination strategies to generate superior trajectories.

**Crossover** We identify high-performing trajectory segments across different paths and combine them to create hybrid trajectories that inherit the strengths of multiple parents:

$$t_{new}^{cross} = \text{Crossover}(t_i, t_j, \alpha),$$

where $\alpha$ determines the crossover points and combination strategy.

**Transfer Learning** Knowledge and strategies from successful trajectories are systematically transferred to enhance less developed paths:

$$t_{new}^{transfer} = \text{Transfer}(t_i, \{t_j, t_k, ...\}, \beta),$$

where $\beta$ controls the transfer mechanism and knowledge adaptation process.

**Restructuring** Trajectory restructuring based on collective insights from the trajectory pool:

$$t_{new}^{restructure} = \text{Restructure}(\mathcal{T}_i, \gamma),$$

where $\gamma$ guides the restructuring process using global trajectory analysis.

### 4.4 Refinement Operation

The refinement phase represents the culmination of our self-evolution approach, focusing on trajectory optimization and final selection based on comprehensive evaluation metrics.

#### 4.4.1 Evaluation Function

To effectively guide the evolutionary process and select optimal trajectories, we design a multi-dimensional reward function that evaluates trajectory quality across several critical axes:

$$\text{Reward}(t, T) = \alpha \cdot \text{TaskCompletion}(t, T) + \beta \cdot \text{ReasoningQuality}(t) + \gamma \cdot \text{Efficiency}(t),$$

where $\text{TaskCompletion}(t, T)$ measures how effectively trajectory $t$ solves task $T$ through structural validation (e.g., non-empty patch files, sufficient code-editing steps, reasonable trajectory length), $\text{ReasoningQuality}(t)$ evaluates the logical coherence, depth and robustness of the reasoning process, and $\text{Efficiency}(t)$ quantifies computational efficiency in terms of reasoning steps and resource utilization. The hyperparameters $\alpha$, $\beta$, $\gamma$, and $\delta$ control the relative importance of each evaluation dimension and can be adjusted based on task requirements.

We implement this reward function using a combination of automatic metrics and specialized evaluators that analyze both the process and outcome of each trajectory:

$$\text{TaskCompletion}(t, T) = \text{AutoEval}(t, T) + \lambda \cdot \text{ExpertEval}(t, T),$$

where AutoEval consists of rule-based structural validation metrics, while ExpertEval incorporates LLM-based evaluation of solution quality.

Table 1: Performance comparison of our proposed SE-Agent and other frameworks on SWE-bench Verified, evaluated with Pass@1 and Pass@5 across various LLMs. SWE-Agent is a CodeAct-based framework, and SWE-Search is MCTS-based. The best results are highlighted in **bold**. 🧑 indicates open-source LLMs, while 🔒 indicates closed-source LLMs.

| LLM | SWE-Agent | | SWE-Search (MCTS) | | SE-Agent (Ours) | |
|---|---|---|---|---|---|---|
| | **Pass@1** | **Pass@5** | **Pass@1** | **Pass@5** | **Pass@1** | **Pass@5** |
| DeepSeek-V3-0324 🧑 | 31.6% | 35.8% | 39.4% | 41.8% | **54.8%** | **58.4%** |
| Qwen-2.5-72b-Instruct 🧑 | 18.8% | 20.6% | 23.4% | 26.2% | **38.8%** | **42.4%** |
| Llama-3.1-70b-Instruct 🧑 | 15.4% | 17.8% | 21.8% | 23.6% | **32.6%** | **35.2%** |
| GPT-4o 🔒 | 22.4% | 25.4% | 32.6% | 35.8% | **40.4%** | **44.8%** |
| Claude-3.7-Sonnet 🔒 | 40.6% | 43.2% | 47.4% | 50.6% | **61.2%** | **63.6%** |

### 4.4.2  Selection and Convergence

Building upon our comprehensive evaluation function, we implement a strategic selection mechanism that balances trajectory quality and diversity to drive the evolutionary process forward:

$$\mathcal{T}_{i+1} = \text{Select}(\mathcal{T}_i \cup \mathcal{T}_i' \cup \mathcal{T}_i^{\text{new}}, k),$$

where $k$ is the number of elite trajectories to maintain. The mechanism employs a hybrid approach to automatically retain the top-performing trajectories based on reward scores. Meanwhile, it ensures representation of distinct reasoning approaches by calculating trajectory dissimilarity metrics.

This selection process continues iteratively until either a predefined number of evolution cycles is completed or convergence criteria are met (e.g., when the improvement in maximum reward falls below a threshold $\epsilon$ for consecutive iterations). The final output is the highest-rewarding trajectory:

$$t^* = \arg\max_{t \in \mathcal{T}_f} \text{Reward}(t, T),$$

where $\mathcal{T}_f$ is the final trajectory pool after all evolution cycles.

This refined selection and convergence approach embodies the essence of "collective competition and genetic emergence," where trajectories compete and collaborate through structured evolution. Through this mechanism, SE-Agent achieves two critical advantages: (1) exploring substantially larger solution spaces by systematically navigating beyond local optima and (2) leveraging cross-trajectory inspiration to efficiently enhance performance while minimizing the impact of suboptimal reasoning paths. These advantages enable SE-Agent to tackle complex multi-step reasoning tasks with unprecedented effectiveness and efficiency, demonstrating the power of self-evolution.

## 5  Experiments

### 5.1  Experimental Setup

**Benchmark**    In our experiments, we utilize SWE-bench Verified, a curated subset of the broader SWE-bench, consisting of 500 real-world GitHub issues. This benchmark is meticulously designed to provide a self-contained and controlled environment for evaluating framework performance, with a specific focus on functional bug fixes. Each instance in the benchmark includes a natural language description of a GitHub issue and its corresponding code repository, serving as the sole input to the model. To ensure the rigor of evaluation, developer-written unit tests are employed to verify the correctness of model-generated patches. This combination of real-world scenarios and systematic validation establishes SWE-bench Verified as a robust and consistent benchmark for assessing the effectiveness of automated bug-fixing systems.

**Evaluation Metrics**    To evaluate the performance of our proposed SE-Agent, we employ two key metrics, i.e., the resolve rate (Pass@1) and Pass@5. Pass@1 quantifies the percentage of issues

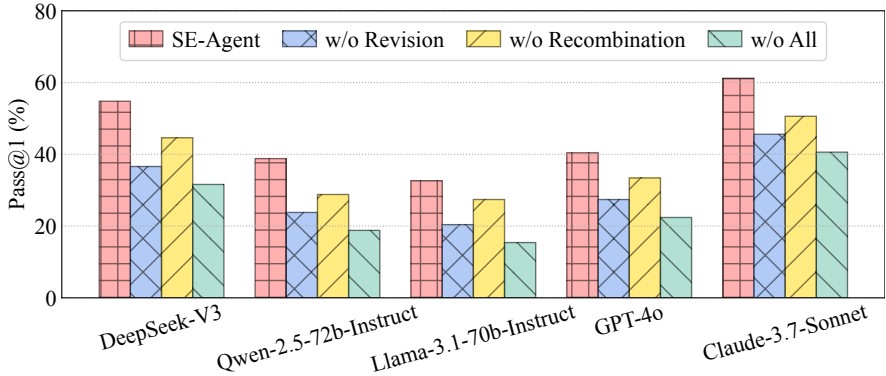

Figure 2: Ablation study of SE-Agent on SWE-bench Verified with three variants.

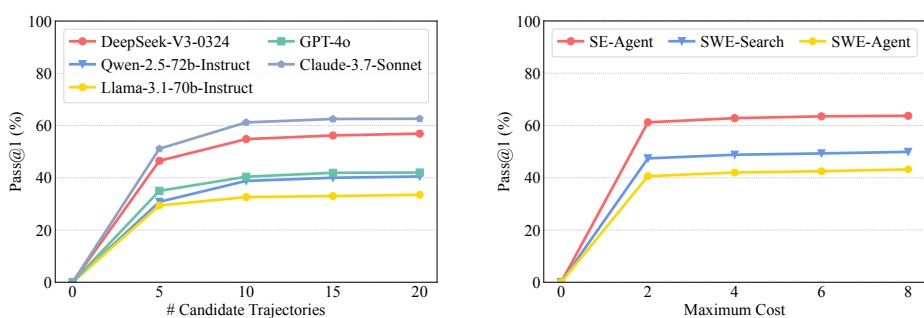

Figure 4: Performance of SE-Agent with different numbers of candidate trajectories and different maximum costs of API on SWE-bench Verified.

successfully resolved on the first attempt, serving as an indicator of the system's overall effectiveness in generating accurate solutions without requiring multiple iterations. In contrast, Pass@5 assesses the percentage of issues for which a correct solution is identified within five attempts, providing insight into the agent's search efficiency under constrained iteration budgets. Together, these metrics offer a comprehensive evaluation framework, capturing both the precision of the agent's initial predictions and its ability to explore solution spaces efficiently.

**Baselines** For a comprehensive and fair evaluation, we compare the performance of SE-Agent against two widely recognized baselines: SWE-Agent (CodeAct-based) and SWE-Search (MCTS-based). These baselines represent high-performing, open-source frameworks frequently utilized in recent research on automated software engineering tasks. The comparison is conducted across multiple models, encompassing both open-source and closed-source paradigms. Specifically, we evaluate three leading open-source models (DeepSeek-V3-0324, Qwen-2.5-72b-Instruct, and Llama-3.1-70b-Instruct) as well as two state-of-the-art closed-source models (GPT-4o and Claude-3.7-Sonnet). Notably, SE-Agent can be integrated into the existing framework as a plug-and-play module. Here we use SWE-Agent as the basis for subsequent experiments.

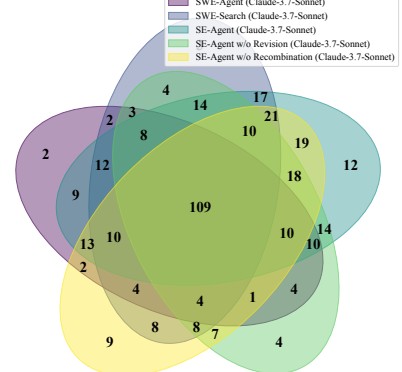

Figure 3: Venn diagram of resolved issues on SWE-bench Verified.

## 5.2 Experimental Results

**Performance Comparison** Table 1 presents a performance comparison between our proposed SE-Agent and existing frameworks on SWE-bench Verified. The results indicate that SE-Agent consistently outper-

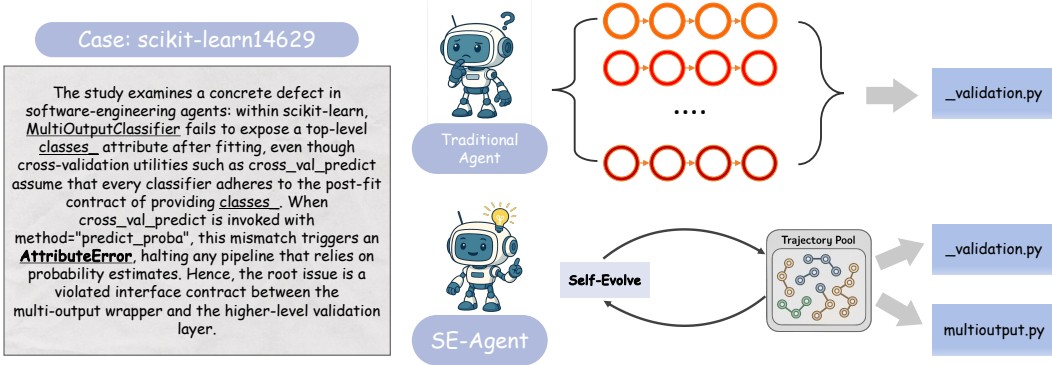

Figure 5: **SWE-bench case SCIKIT-LEARN #14629.** *Top (Traditional Agent).* Search paths are highly homogeneous: each rollout edits `_validation.py`, yielding near-duplicate "quick-fix" patches that hide the visible error yet fail hidden tests. *Bottom (SE-Agent).* By mixing and recombining whole trajectories, our agent explores diverse regions of the patch space, discovers `multioutput.py`, and adds a one-line write of `classes_`, providing a root-level repair that passes the full test suite.

forms the baselines across all five evaluated LLMs. Compared with SWE-Agent baseline, SE-Agent delivers a relative improvement of +112% (Llama-3.1-70B), +80% (GPT-4o) and +51% (Claude-3.7-Sonnet). Against the stronger MCTS-based SWE-Search, the relative gains are still +30% on average. Notably, all five models demonstrate substantial and consistent performance gains when integrated with our proposed framework, highlighting the generalizability and effectiveness of SE-Agent across diverse model families.

**Ablation Study** In this part, we conduct the ablation study to explore the contribution of each designed module in SE-Agent. Therefore, we compare SE-Agent with three different variants: (i) w/o Revision, i.e., the Revision operation is removed, resulting in only multiple homogenized trajectories. (ii) w/o Recombination, where we do not use the Recombination operation for trajectory interaction. (iii) w/o All, which does not use any trajectory optimization operation. The results are presented in Figure 2. These results illustrate two facts: (i) All designed modules are important for SE-Agent. If any module is removed, Pass@1 will decrease. (ii) Revision is effective for the performance enhancement of SE-Agent because it provides a diverse set of trajectories for subsequent Recombination. As illustrated in Figure 3, we further conduct a detailed analysis of the overlap in successfully resolved issue instances across different frameworks on SWE-bench Verified, using a Venn diagram for visualization. The results reveal that our proposed SE-Agent uniquely solves 12 issue instances that none of the other models are able to address. In addition, SE-Agent exhibits substantial overlap with leading baselines in the set of resolved issues, further underscoring its competitive overall performance. This analysis highlights two key advantages of SE-Agent: its competitive effectiveness in solving tasks tackled by state-of-the-art models and its distinct capability to address a broader range of difficult or previously unsolved issues, demonstrating robustness and complementary problem-solving strength.

In Figure 4, we investigate the effect of two key hyperparameters on the performance of SE-Agent: the number of candidate trajectories and the maximum API cost. Results show SE-Agent reaches near-optimal performance with just 10 candidate trajectories, demonstrating our trajectory-based search strategy's efficiency through inter-trajectory interactions. The maximum API cost reflects the depth of SE-Agent's exploration. Under the same cost budgets, SE-Agent consistently outperforms baseline methods in Pass@1 scores, validating our self-evolution framework's effectiveness.

**Case Study** To better illustrate the concrete implementation of our method, Figure 6 provides a complete case study demonstrating how SE-Agent progressively optimizes trajectories through its three core operations. As shown in Figure 5, the crash surfaces inside `_validation.py`, yet the root cause is that the wrapper in `multioutput.py` never stores the required `classes_` field after training. Traditional ReAct/MCTS agents cling to the stack trace: (i) they pose the bug too narrowly, (ii) next-token prediction keeps every edit local, and (iii) their roll-outs are near-identical, so each

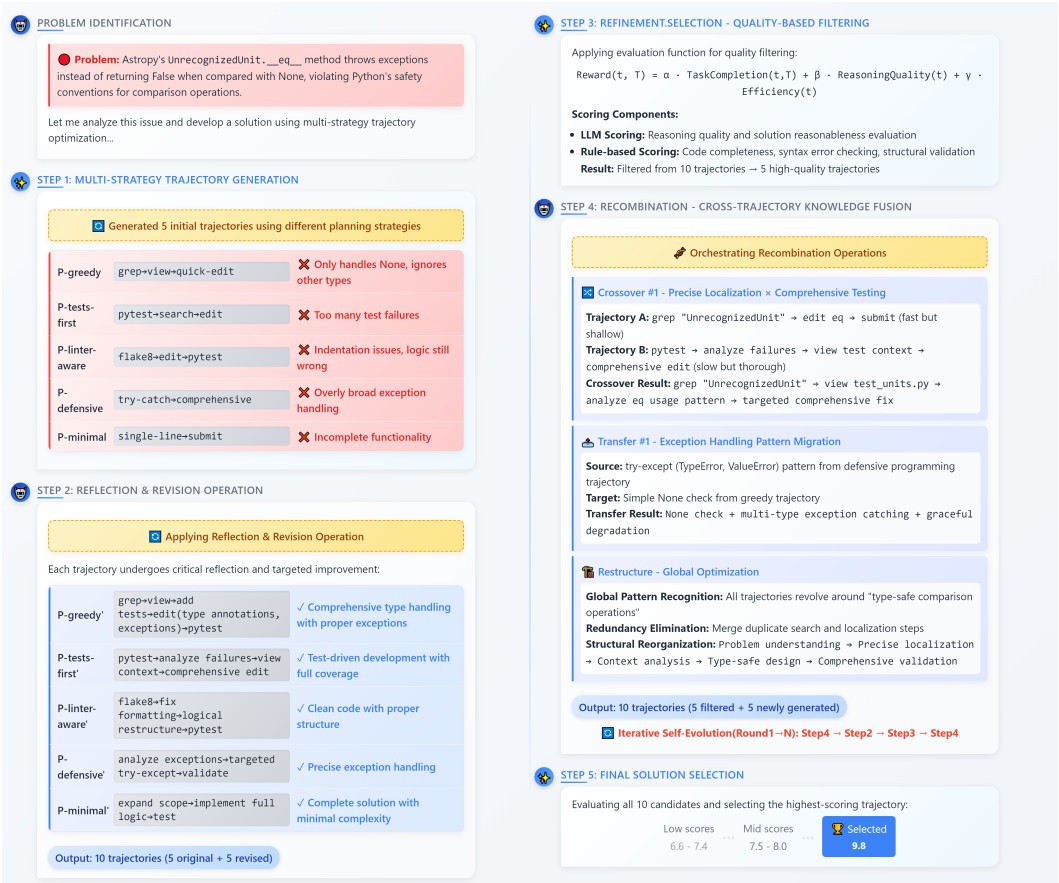

Figure 6: To better illustrate the concrete implementation of our method, we provide a comprehensive case study that demonstrates how SE-Agent progressively optimizes trajectories through its three core operations.

patch merely tweaks `_validation.py`. Our SE-Agent sidesteps this tunnel vision by iteratively *interacting with and evolving entire trajectories*; this trajectory-level evolution serves as an implicit regularizer, forcing the search to generate genuinely novel solutions rather than minor variants of the same fix. Figure 7 in the Appendix provides a detailed comparison of the trajectories output by SE-Agent before and after optimization. Notably, none of the top three public frameworks on SWE-bench can solve this case.

# 6  Conclusion

In this work, we introduced SE-Agent, a self-evolution framework designed to enhance the multi-step reasoning capabilities of LLM-based agents through iterative trajectory optimization. By revisiting, recombining, and refining previously generated trajectories, SE-Agent systematically expands the exploration space and leverages cross-trajectory insights to improve decision-making efficiency. Experimental evaluations on SWE-bench Verified demonstrate that SE-Agent consistently outperforms strong baselines across multiple LLMs. Our findings highlight the value of incorporating self-evolutionary principles into agent design, paving the way for more robust and adaptable reasoning frameworks in complex environments. Looking forward, we aim to extend the self-evolution paradigm of SE-Agent to a wider spectrum of path-search problems—including iterative search-reason frameworks such as DeepSearch, reinforcement-learning policy discovery, and embodied-intelligence scenarios where agents must reason and act in the physical world.

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

# A    Implementation Details

To ensure a fair comparison, we adopt identical prompt formats across all models evaluated in this paper. In our proposed SE-Agent framework, we set the number of candidate trajectories to 10 by default, striking a balance between exploration diversity and computational efficiency.

We begin by employing five distinct planning strategies, as described in Section B.1.1, to generate a diverse set of initial trajectories that reflect varied reasoning patterns. These serve as the foundational seeds for further optimization. Next, we apply the reflection and revision operations (Section B.1.2) to each initial trajectory. This process generates up to ten revised trajectories by encouraging self-criticism and iterative improvement within the agent's reasoning process.

Following the generation of candidate trajectories, we perform the recombination operation (Section B.2), which enables the agent to integrate complementary insights from different trajectories, promoting information fusion and enhancing reasoning coherence. Subsequently, the refinement operation (Section B.3) is applied to finalize the optimized trajectories, ensuring logical consistency and code validity when applicable.

For deployment, we run all open-source models locally, including DeepSeek-V3-0324, Qwen-2.5-72B-Instruct, and LLaMA-3.1-70B-Instruct, using NVIDIA A100 GPUs with 80GB of memory. For closed-source models such as GPT-4o and Claude-3.7-Sonnet, we access them via the official APIs provided by OpenAI and Anthropic, respectively. All experiments are conducted under the same evaluation setting on SWE-bench Verified to ensure consistency and reproducibility.

# B    Prompt

## B.1    Revision Operation

### B.1.1    Multi-Planning

**Planning-1**

STRATEGY:
1. Always start by trying to replicate the bug that the issues discusses.
If the issue includes code for reproducing the bug, we recommend that you re-implement that in your environment, and run it to make sure you can reproduce the bug.
Then start trying to fix it.
If the bug reproduction script does not print anything when it successfully runs, we recommend adding a print("Script completed successfully, no errors.") command at the end of the file,
so that you can be sure that the script indeed ran fine all the way through.
2. Locate relevant code using the find and search commands. 'open' the file you want to edit.
I highly recommend using 'fileshow' command before you locate the relevant code to get information about the folder where the target code file resides, this can help you think about what code to locate and modify to better solve the problem.
3. Use the 'edit' command to perform edits.
4. When you think you've fixed the bug, re-run the bug reproduction script to make sure that the bug has indeed been fixed.
5. Create additional tests to verify the fix in a style similar to the existing reproduction script. In particular, make sure to test edge cases.
If you find any issues, go back to the file you edited and perform further edits.

**Planning-2**

STRATEGY:
1. **Reproduce the Issue:** - Execute the provided test case or script to confirm the presence of the bug. If the script lacks output verification, insert a validation statement (e.g., 'assert "Expected Behavior" in output') to ensure full execution.

2. **Code Investigation:** - Trace the execution flow using logging or debugging tools. Identify the exact module, function, or line causing the discrepancy. Cross-reference with documentation or API contracts if needed.
3. **Modification Process:** - Apply changes incrementally using version-controlled edits. Annotate modifications with comments explaining the rationale behind each adjustment (e.g., "Fix: Resolved race condition by adding mutex lock").
4. **Validation:** - Re-run the original test alongside regression tests covering related functionality. Include boundary conditions (e.g., empty inputs, extreme values) and parallel execution scenarios if applicable.
5. **Documentation:** - Update changelogs or inline documentation to reflect the fix. If the issue reveals a broader architectural weakness, propose a follow-up task to address systemic improvements.

## Planning-3

STRATEGY:
1. **Reproduce the Issue:** - Execute the provided bug reproduction script or steps in your local environment. - If the script runs silently, insert a confirmation message (e.g., 'print("Reproduction successful—bug observed.")') to verify execution.
2. **Code Investigation:** - Identify the relevant code sections using search tools or IDE navigation. - Open the suspected files and analyze the logic around the reported issue.
3. **Implement Fixes:** - Use an editor to modify the problematic code, ensuring changes align with the intended behavior. - Document adjustments with inline comments explaining the rationale.
4. **Verify the Fix:** - Re-run the reproduction script to confirm the bug is resolved. - Check for regression by ensuring existing functionality remains intact.
5. **Expand Test Coverage:** - Develop new test cases, including edge scenarios, to validate robustness. - If failures occur, refine the fix and repeat verification until all tests pass.
6. **Final Review:** - Cross-check changes against project coding standards. - Summarize modifications and test results for documentation or version control.

## Planning-4

STRATEGY:
1. Begin by thoroughly analyzing the reported issue to understand its context and expected behavior. If the issue provides a minimal reproducible example, execute it in your environment to confirm the bug. Modify the script to include a clear success/failure indicator (e.g., 'print("Validation passed.")') to ensure full execution.
2. Systematically trace the codebase using grep, IDE search, or debugging tools to pinpoint the exact file(s) and logic responsible for the unexpected behavior. Open the relevant files for inspection.
3. Make targeted adjustments using your preferred editor or IDE, ensuring changes align with the project's architecture. Document modifications with inline comments explaining the rationale.
4. Validate the fix by re-running the original reproduction script and checking for resolved behavior. Confirm no regressions occur in related functionality.
5. Expand test coverage by designing new test cases that stress edge conditions, boundary values, and atypical inputs. Integrate these into the existing test suite. If failures persist, iterate on the fix and retest until all scenarios pass.
6. Finally, review the changes for code style consistency and potential optimizations before submission.

**Planning-5**

STRATEGY:
1. **Reproduce the Issue:** Begin by carefully executing the steps or code provided in the bug report to confirm the problem. If the reproduction script runs silently, insert a confirmation message (e.g., 'print("Verification: Initial bug reproduction successful.")') to ensure full execution.
2. **Code Investigation:** Systematically trace the issue using search tools or debugging utilities. Identify the exact file(s) and section(s) involved, then open them for analysis.
3. **Implement Changes:** Use precise editing commands to modify the problematic code. Document each adjustment to track potential impacts.
4. **Validation:** Re-run the reproduction script after each edit to verify the fix. If the issue persists, refine the changes incrementally.
5. **Regression Testing:** Expand test coverage by designing new test cases that mirror the original issue's context, including edge scenarios. Iterate if failures occur.
6. **Final Confirmation:** Ensure the fix doesn't introduce new issues by running the full test suite or related workflows. Only conclude when all validations pass.

### B.1.2 Reflection and Revision

**Reflection-Find the Critical Step**

You are an expert in analyzing agent trajectories. Your task is to identify the single most influential critical decision point in the trajectory. A critical decision point is the step that has the most significant impact on the entire solution, typically:
1. Finding the core location of the problem (such as locating specific files or functions)
2. Discovering the breakthrough point for the solution 3. Making key modifications or judgments 4. Determining the correct execution path
You need to analyze the entire trajectory, identify the single most critical step, and provide: - Step number - Why this step is the most critical - The impact of this step on the final solution
Output must strictly follow JSON format, containing: - `critical_step` object with fields: step, action, reasoning, impact
The following example demonstrates the expected output format and level of detail for critical point identification:
## Input Example
```json "trajectory": [ "step": 1, "action": "Search for _check_list_display_item function in src directory", "observation": "No matches found" , "step": 2, "action": "Execute ls -F to view file structure", "observation": "Listed directory structure" , "step": 3, "action": "Search for _check_list_display_item function in django directory", "observation": "Found 2 matches in django/contrib/admin/checks.py" , "step": 4, "action": "Open django/contrib/admin/checks.py file", "observation": "Successfully opened, 1116 lines total" , "step": 5, "action": "Search for _check_list_display_item function in checks.py", "observation": "Found at lines 714 and 718" , "step": 6, "action": "Navigate to line 718 to view function definition", "observation": "Displayed complete implementation of _check_list_display_item function" , "step": 7, "action": "Modify function logic to resolve admin.E108 error", "observation": "Successfully edited code" , "step": 8, "action": "Submit solution", "observation": "Task completed" ] ``` ``` ## Output Example "critical_step": "step": 3, "action": "Search for _check_list_display_item function in django directory", "reasoning": "This is the most critical breakthrough point in the entire solution process. After the first two search attempts failed, this step correctly located the exact position of the problem function, transitioning from a directionless state to having a clear modification target. Without this step, all subsequent operations would be impossible.", "impact": "Determined whether the entire task could be successfully completed. Finding the function location is a prerequisite for solving the problem and marks the transition from exploration phase to implementation phase." ```

## Reflection-Conclude Feature

You are an AI assistant specialized in analyzing code patches. I will provide a GitHub issue (`problem_statement`) and a corresponding patch. Your task is to analyze this patch and provide detailed insights that could help develop an alternative solution.

Follow these steps:

1. Analyze the patch file and understand the changes made
2. Determine the core methods and techniques used to solve the problem
3. Identify the main files and sections that were modified
4. Identify key assumptions and limitations in the current solution

Return your analysis in JSON format with the following fields:

- `approach_summary`: Summary of the main approach used in the first solution
- `modified_files`: List of files that were modified
- `key_changes`: Description of key code changes in the patch
- strategy: The core solution strategy at an abstract level
- `specific_technique_from_first_solution`: Specific technique used that should be avoided in alternative solutions
- `specific_files_or_functions`: Files or functions that should not be modified in the same way
- `assumptions_made_in_first_solution`: Assumptions made in the first solution
- `component_not_touched_in_first_solution`: Components or key functions not touched but potentially relevant
- `different_perspective`: A different perspective for looking at the problem

The following examples are provided only for reference to illustrate the expected level of detail and abstraction for each field. Your analysis should be based on your own understanding of the patch and problem:

`approach_summary` example: "Added a conditional check to handle MultiOutputClassifier by accessing classes through the `estimators_` attribute"

`modified_files` example: ["sklearn/model_selection/_validation.py"]

`key_changes` example: "Added a condition to check if estimator has 'estimators_' attribute, then uses `estimator.estimators_[i_label].classes_` instead of `estimator.classes_[i_label]` for MultiOutputClassifier"

strategy example: "Component-specific exception handling" (instead of "Interface extension to provide unified attribute access")

`specific_technique_from_firs_solution` example: "Direct attribute checking with hasattr() and conditional branching"

`specific_files_or_functions` example: "`_fit_and_predict` function in sklearn/model_selection/_validation.py"

`assumptions_made_in_first_solution` example: "Assumes that only MultiOutputClassifier needs special handling for `classes_` attribute access"

`component_not_touched_in_first_solution` example: "MultiOutputClassifier class in sklearn/multioutput.py which could implement `classes_` attribute directly"

`different_perspective` example: "API consistency perspective: make MultiOutputClassifier conform to the same interface as other classifiers instead of modifying the validation module"

Problem: `problem_statement` Trajectory: trajectory Patch: `model_patch`

## Revision

Let's develop a different approach to fix it.

I've analyzed the first solution attempt, and here's what I found:

The first solution approached this problem by `approach_summary`. It modified `modified_files`, and the key changes involved `key_changes`.

The core strategy used was "strategy" which may have limitations. Specifically, the solution used `specific_technique` and focused on changing `specific_files`.

This approach makes some assumptions: assumptions

For our alternative solution, I want you to explore a completely different approach. Instead of following the same strategy, consider looking at this from a `different_perspective` angle.
You might want to investigate `component_not_touched` as a potential area for your solution.
Remember, your goal is to create a patch that:
1. Solves the same problem but uses a fundamentally different approach
2. Avoids the techniques used in the first solution
3. Challenges the assumptions made in the first solution
4. Still aims to generate a patch that passes all tests, and use the 'submit' command when you believe your modifications are complete
Please analyze the problem again from this new perspective and develop your alternative solution.

## B.2 Recombination Operation

**CrossOver**

You are an expert in analyzing and synthesizing agent trajectories. Your task is to critically analyze two different trajectories and create a new optimized trajectory by combining the best elements from both approaches.
Your fusion process should: 1. Identify the strengths and weaknesses of each trajectory 2. Extract the most effective strategies and techniques from both 3. Creatively integrate these elements into a coherent new trajectory 4. Ensure the new trajectory maintains logical flow and consistency 5. Avoid simply concatenating the trajectories - create genuine synthesis
You need to analyze both trajectories and provide: - Analysis of strengths from trajectory A - Analysis of strengths from trajectory B - A new fused trajectory that combines the best aspects - Rationale for the fusion decisions
Output must strictly follow JSON format, containing: - `trajectory_a_strengths`: list of strengths from first trajectory - `trajectory_b_strengths`: list of strengths from second trajectory - `fused_trajectory`: new trajectory steps combining best elements - `fusion_rationale`: explanation of fusion decisions
Trajectory A: `trajectory_a`
Trajectory B: `trajectory_b`
The following example demonstrates the expected output format and level of detail for trajectory fusion:
`## Input Example` Trajectory A: "trajectory": [ "step": 1, "action": "`search_dir` for target function", "observation": "found function location quickly" , "step": 2, "action": "directly edit function", "observation": "made changes without full analysis" , "step": 3, "action": "submit solution", "observation": "task completed" ]
Trajectory B: "trajectory": [ "step": 1, "action": "analyze problem statement thoroughly", "observation": "understood the root cause" , "step": 2, "action": "`search_file` to find exact location", "observation": "took longer but found precise location" , "step": 3, "action": "review existing code logic", "observation": "identified potential side effects" , "step": 4, "action": "implement careful modification", "observation": "made robust changes" , "step": 5, "action": "test the solution", "observation": "verified correctness" , "step": 6, "action": "submit solution", "observation": "task completed successfully" ]
`## Output Example` "trajectory_a_strengths": [ "Efficient search strategy using `search_dir`", "Quick execution with minimal steps", "Direct approach to problem solving" ], "trajectory_b_strengths": [ "Thorough problem analysis at the beginning", "Careful code review to identify potential issues", "Testing phase to ensure solution robustness", "More comprehensive approach to code modification" ], "fused_trajectory": [ "step": 1, "action": "analyze problem statement to understand root cause", "reasoning": "Combined trajectory B's thorough analysis with trajectory A's efficiency goal" , "step": 2, "action": "`search_dir` for target function with precise search terms", "reasoning": "Used trajectory A's efficient search method with trajectory B's precision approach" , "step": 3, "action": "review existing code logic and identify modifications needed", "reasoning": "Incorporated

trajectory B's careful review step before making changes" , "step": 4, "action": "implement modification with consideration for edge cases", "reasoning": "Combined trajectory A's direct editing with trajectory B's careful consideration" , "step": 5, "action": "submit solution after quick validation", "reasoning": "Balanced trajectory A's efficiency with trajectory B's testing approach" ], `fusion_rationale`: "The fused trajectory combines trajectory A's efficiency and direct approach with trajectory B's thoroughness and careful analysis. It maintains a streamlined execution path while incorporating critical analysis and validation steps, resulting in a solution that is both efficient and robust."

## Transfer

You are an expert in optimizing agent trajectories through transfer learning. Your task is to enhance a target trajectory by transferring effective strategies, insights, and approaches from a pool of reference trajectories.

Your transfer learning process should: 1. Analyze the target trajectory to identify areas for improvement 2. Extract valuable patterns, strategies and techniques from the reference trajectories 3. Transfer these elements to enhance the target trajectory 4. Ensure the enhanced trajectory maintains logical coherence and consistency 5. Focus on meaningful knowledge transfer, not simply adding steps

Target Trajectory: `trajectory_target`

Reference Trajectory Pool: `traj_pool`

Carefully analyze both the target trajectory and reference pool, then create an enhanced version of the target trajectory that incorporates the most valuable elements from the reference trajectories.

Your output should be a single JSON object representing the enhanced trajectory, following this exact format: "trajectory": [ "step": 1, "action": "action description", "observation": "observation description" , "step": 2, "action": "action description", "observation": "observation description" , ... ]

Make sure the enhanced trajectory: - Addresses weaknesses in the original target trajectory - Incorporates valuable insights from reference trajectories - Maintains a coherent problem-solving approach - Includes specific implementation details - Has logical progression between steps - Is complete enough to solve the task effectively

## Restructure

You are an expert in large-scale trajectory restructuring. Your task is to synthesize a new reasoning trajectory by analyzing the global structure of a trajectory population. Unlike `Crossover` or `Transfer` that focus on local segment manipulation, this task requires holistic restructuring based on global insights across all input trajectories.

Your restructuring process should: 1. Analyze the entire trajectory pool to discover abstract patterns, common subgoals, and shared structures 2. Identify redundant reasoning paths and filter out ineffective or repetitive steps 3. Synthesize a completely new trajectory that aligns with the overall problem-solving objective, but reflects a novel and optimized reasoning process 4. Maintain logical consistency, completeness, and step-wise progression of the trajectory

Trajectory Pool: `trajectory_pool`

You must generate a single restructured trajectory that combines the collective strengths and high-level reasoning strategies inferred from the input trajectories. Your output should be a single JSON object representing the newly restructured trajectory, following this exact format:

```
{
  "new_trajectory": [
    {
      "step": 1,
      "action": "action description",
      "observation": "observation description",
```

```
      "reasoning": "why this step was chosen and how it contributes"
    },
    {
      "step": 2,
      "action": "action description",
      "observation": "observation description",
      "reasoning": "rationale for this step"
    }
    ...
  ]
}
```

Make sure the restructured trajectory: - Reflects global insights derived from the trajectory pool - Avoids redundancy and overly local reasoning - Introduces a coherent and efficient solution strategy - Demonstrates abstract synthesis and long-range planning - Forms a complete and executable path to solve the task

## B.3 Refinement Operation

### B.3.1 Evaluation

**TaskCompletion**

You are an expert in large-scale trajectory restructuring. Your task is to synthesize a new reasoning trajectory by analyzing the global structure of a trajectory population. Unlike `Crossover` or `Transfer` that focus on local segment manipulation, this task requires holistic restructuring based on global insights across all input trajectories.

Your restructuring process should: 1. Analyze the entire trajectory pool to discover abstract patterns, common subgoals, and shared structures 2. Identify redundant reasoning paths and filter out ineffective or repetitive steps 3. Synthesize a completely new trajectory that aligns with the overall problem-solving objective, but reflects a novel and optimized reasoning process 4. Maintain logical consistency, completeness, and step-wise progression of the trajectory

Trajectory Pool: `trajectory_pool`

You must generate a single restructured trajectory that combines the collective strengths and high-level reasoning strategies inferred from the input trajectories. Your output should be a single JSON object representing the newly restructured trajectory, following this exact format:

```
{
  "new_trajectory": [
    {
      "step": 1,
      "action": "action description",
      "observation": "observation description",
      "reasoning": "why this step was chosen and how it contributes"
    },
    {
      "step": 2,
      "action": "action description",
      "observation": "observation description",
      "reasoning": "rationale for this step"
    }
    ...
  ]
}
```

Make sure the restructured trajectory: - Reflects global insights derived from the trajectory pool - Avoids redundancy and overly local reasoning - Introduces a coherent and efficient so-

lution strategy - Demonstrates abstract synthesis and long-range planning - Forms a complete and executable path to solve the task

# C  Case Study

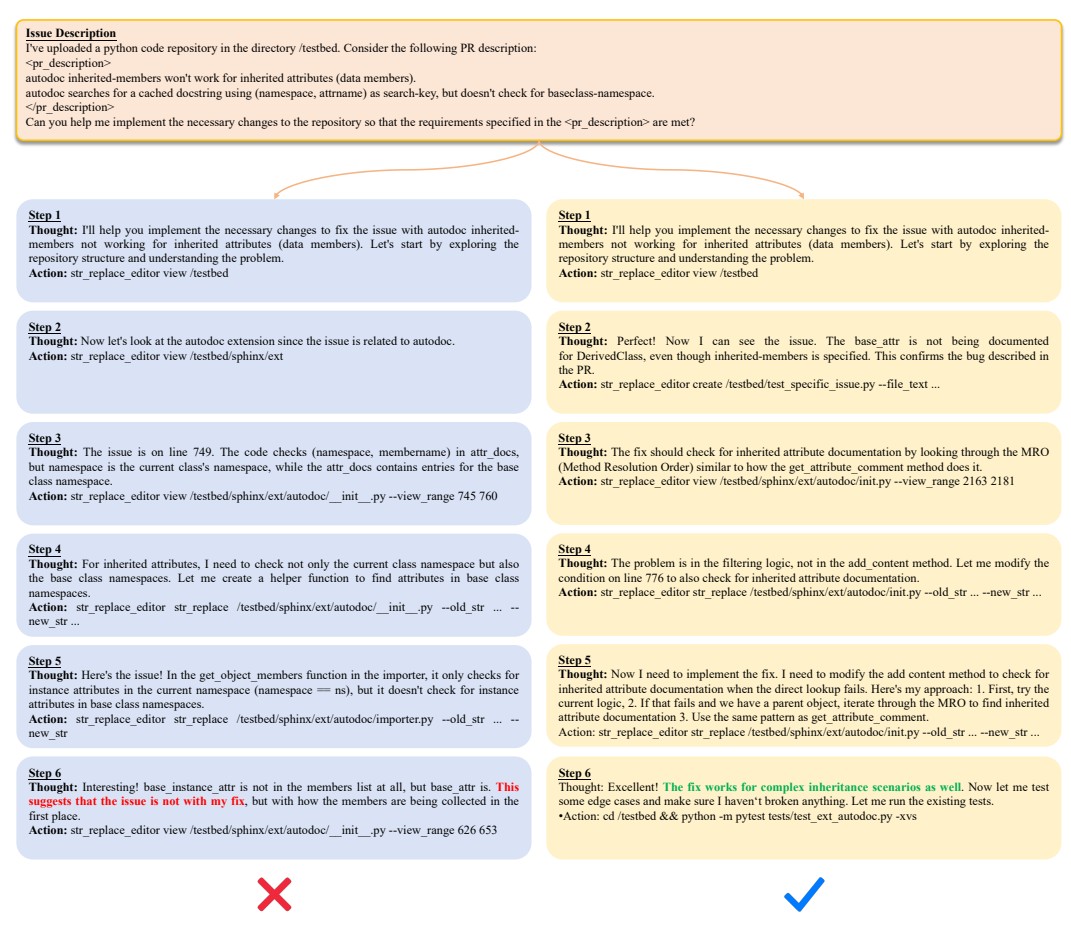

Figure 7: A case analysis comparing the trajectories output by SE-Agent before and after optimization.

