# OpenReview forum: "SE-Agent: Self-Evolution Trajectory Optimization in Multi-Step Reasoning with LLM-Based Agents"
_NeurIPS.cc/2025/Conference — NeurIPS 2025 poster_

### Official Review · Reviewer_aGqC · 2025-07-01

**Clarity:** 2
**Significance:** 2
**Originality:** 2
**Rating:** 4
**Confidence:** 4

**Summary:**

This paper presents SE-Agent, a framework that enhances LLM agent performance by treating their reasoning paths as "trajectories" for optimization. The system employs evolutionary operation to iteratively improve a pool of trajectories and discover superior solutions. Validated on the SWE-bench benchmark, the method demonstrates significant performance improvements over baseline models.

**Questions:**

See weaknesses

**Ethical Concerns:**

["NO or VERY MINOR ethics concerns only"]

**Final Justification:**

The authors have provided a comprehensive and thoughtful rebuttal that addresses all of my initial concerns. Overall, the paper makes a solid contribution to the area of prompt engineering for LLM agents.

That said, my overall score remains moderate due to the inherent limitations of the proposed approach. Specifically, while trajectory-based prompt engineering offers clear short-term gains, its generalizability and scalability are less convincing. The method requires relearning when applied to new domains beyond software engineering, which may be both costly and resource-intensive. Furthermore, the paper does not evaluate the transferability of the approach to tasks outside the coding domain, and comparative experiments on broader baselines are missing. These gaps limit the demonstrated impact and applicability of the method.

**Limitations:**

Yes

**Quality:**

2

**Strengths And Weaknesses:**

Strengths:
1. The empirical results are strong, showing substantial relative improvements over established baselines on the challenging SWE-bench benchmark. The method appears to generalize well across multiple open-source and closed-source LLMs.

Weaknesses:
1. The idea of collecting a pool of trajectories or experiences to guide or improve subsequent reasoning attempts is a well-established concept in agent learning, seen in frameworks like ExpeL[1] and AvaTar[2], which also learns from diverse, past experiences to improve agent performance. The "self-evolution" framing appears to be a new wrapper for a familiar concept, and its distinction from prior work in experience-based improvement or iterative refinement is not sufficiently articulated.
2. The empirical evaluation, while strong, is narrowly focused on a single, highly specific benchmark: SWE-bench.  This makes it difficult to assess the generality of the SE-Agent framework. Results across a wider range of tasks are needed to demonstrate that this isn't just a bespoke solution for fixing GitHub issues. Its applicability to other code-related benchmarks like TerminalBench for terminal coding, LiveCodeBench for competitive programming.
3. The paper lacks the concrete, low-level details needed for a deep understanding and confident replication of the method. The descriptions of the Revision, Recombination, and Refinement operators remain at a very high level. For example, how exactly does the
Crossover function identify "high-performing trajectory segments" and combine them? What does the output of a Mutate operation actually look like?
4. The case study presented in Figure 6 is too vague to be truly insightful. It claims that traditional agents get stuck editing `_validation.py` while SE-Agent discovers the root cause in `multioutput.py`, but it fails to show how the evolutionary process enabled this discovery. A step-by-step walkthrough is needed to illustrate the point.
5. The proposed method generates an initial pool of ten trajectories and then iteratively evolves them. This process could be expensive in terms of number of API calls, tokens, and wall-clock time. The analysis in Figure 5, which plots performance against "Maximum Cost," is insufficient as the "Cost" unit is not defined. A transparent discussion of its resource overhead compared to baselines is required.

[1] ExpeL: LLM Agents Are Experiential Learners. AAAI 24

[2] AvaTaR: Optimizing LLM Agents for Tool Usage via Contrastive Reasoning. NIPS 24

---

> ### Author Rebuttal · Authors · 2025-07-31
>
> # Response to Reviewer aGqC
>
> Thank you for your insightful feedback and recognition of our **strong empirical results** and substantial improvements over baselines on SWE-bench. We address your concerns as follows:
>
> ## **The contributions of SE-Agent (For W1)**
>
> While leveraging historical experience is part of our approach, SE-Agent's core contribution lies in **expanding the search space through multi-trajectory interaction**, overcoming single-trajectory reasoning limitations. This process corresponds to crossover and mutation operations in genetic evolutionary theory, not merely refining individual trajectories.
>
> ExpeL and AvaTar lack **cross-trajectory interaction** mechanisms. Despite self-reflection, isolated trajectories remain prone to cognitive limitations and local optima.
>
> Our DeepSeek-R1 experiments show smaller models with GRPO performed worse than distillation due to trajectory homogenization. Classic MCTS generates multiple paths but lacks real-time information exchange, resulting in limited search efficiency compared to our trajectory interaction mechanism.
>
> **Compared to MCTS**: SE-Agent introduces **trajectory-level evolutionary mechanisms** enabling information exchange across reasoning paths, providing:
>
> a. **Breaking cognitive boundaries**: Competition and collaboration overcome individual path limitations
> b. **Knowledge transfer**: Successful patterns transfer across trajectories through recombination
>
> This fosters **emergent population-level intelligence**. Our experiments show relative gains up to **112% (Llama-3.1-70B)** and **51% (Claude-3.7-Sonnet)**, demonstrating **true capability evolution** beyond individual trajectory sampling.
>
> ## **The Choice of Benchmark (For W2)**
>
> We focus on **SWE-bench Verified** because it presents significant challenges requiring cross-file bug localization, patch generation, and test validation over real-world repositories. Even top models achieve only 50-60% success. Leading works like **SWE-agent and SWE-search** report main results on this benchmark.
>
> LiveCodeBench emphasizes single-turn generation rather than multi-step interaction. Terminal-Bench aligns well with SE-Agent's capabilities but requires additional engineering work—a promising future direction.
>
> ## **A description of the Operators (For W3&W4)**
>
> We demonstrate our specific operations through the following case:
>
> ### **Case Study: Fixing UnrecognizedUnit.__eq__ in Astropy**
>
> **Problem Description**: Astropy's UnrecognizedUnit.__eq__ method throws exceptions instead of returning False when compared with None, violating Python's safety conventions for comparison operations.
>
> **Step 1: Revision Operation - Multi-Strategy Trajectory Generation**
>
> First, generate 5 initial trajectories using 5 different planning strategies:
>
> | Prompt Variant | Trajectory Summary | Main Issues |
> |----------------|-------------------|-------------|
> | P-greedy | `grep→view→quick-edit` | Only handles None, ignores other types |
> | P-tests-first | `pytest→search→edit` | Too many test failures |
> | P-linter-aware | `flake8→edit→pytest` | Indentation issues, logic still wrong |
> | P-defensive | `try-catch→comprehensive` | Overly broad exception handling |
> | P-minimal | `single-line→submit` | Incomplete functionality |
>
> Then perform reflection and revision operations on each trajectory to generate corresponding improved versions:
>
> **Example - Evolution of P-greedy trajectory:**
> - **Original trajectory issue**: Overly narrow type checking
> - **Revised trajectory**: `grep → view → add comprehensive tests → edit(improve type annotations, add exception handling) → pytest`
>
> **Output**: 10 revised trajectories (original 5 + revised 5) proceed to the next stage.
>
> **Step 2: Refinement.Selection - Quality-Based Trajectory Filtering**
>
> Use evaluation function for quality filtering:
> $$\text{Reward}(t, T) = \alpha \cdot \text{TaskCompletion}(t,T) + \beta \cdot \text{ReasoningQuality}(t) + \gamma \cdot \text{Efficiency}(t)$$
>
> Scoring includes two components:
> - **LLM scoring**: Reasoning quality and solution reasonableness evaluation
> - **Rule-based scoring**: Code completeness, syntax error checking, structural validation
>
> **Result**: Remove low-quality trajectories, retain 5 for the next step.
>
> **Step 3: Recombination Operation - Cross-Trajectory Knowledge Fusion**
>
> Based on task complexity, orchestrate operator combinations to generate 5 new trajectories:
> - **2× Crossover**: Fuse complementary segments from different trajectories
> - **2× Transfer Learning**: Transfer successful patterns to target trajectories
> - **1× Restructure**: Global restructuring optimization
>
> **Specific trajectory fusion examples:**
>
> **Crossover #1 - Precise Localization × Comprehensive Testing:**
> - **Trajectory A**: `grep "UnrecognizedUnit" → edit eq → submit` (fast but shallow)
> - **Trajectory B**: `pytest → analyze failures → view test context → comprehensive edit` (slow but thorough)
> - **Crossover Result**: `grep "UnrecognizedUnit" → view test_units.py → analyze eq usage pattern → targeted comprehensive fix`
>
> **Transfer #1 - Exception Handling Pattern Migration:**
> - **Source**: try-except (TypeError, ValueError) pattern from defensive programming trajectory
> - **Target**: Simple None check from greedy trajectory
> - **Transfer Result**: `None check + multi-type exception catching + graceful degradation`
>
> **Restructure - Global Optimization:**
> - **Global pattern recognition**: All trajectories revolve around the core issue of "type-safe comparison operations"
> - **Redundancy elimination**: Merge duplicate search and localization steps
> - **Structural reorganization**: `Problem understanding → Precise localization → Context analysis → Type-safe design → Comprehensive validation`
>
> **Output**: Extended pool contains 10 trajectories (original 5 + newly generated 5)
>
> **Iterative Self-Evolution Support**
>
> SE-Agent supports multi-round iterative optimization, continuing the evolution process based on the current trajectory pool:
>
> **Round 2 Evolution (simplified representation):**
> - **Step2'**: Reflection ⟨5 trajectories⟩ → Revision ⟨...⟩ → `T2'` (10 trajectories)
> - **Step3'**: Quality-Filtering ⟨T2'⟩ → Select ⟨...⟩ → `T3` (5 trajectories)
> - **Step4'**: Recombination ⟨T3⟩ → [Crossover×2, Transfer×2, Restructure×1] → `T4` (10 trajectories)
>
> **Convergence condition**: Stop when consecutive K rounds of highest reward improvement < ε, or reach preset maximum rounds. This iterative mechanism ensures SE-Agent can continuously optimize until finding high-quality solutions or meeting stop conditions.
>
> **Step 4: Final Solution Selection**
>
> Select the highest-scoring trajectory from candidates as the final solution:
>
> **Core Improvement**: SE-Agent evolved from single-point failure repair to a **type-safe universal solution**, generating entirely new solution methods through **explicit inter-trajectory interactions** (Crossover, Transfer, Restructure), covering all invalid input scenarios.
>
> **Key Innovation**: This trajectory-level collaboration is **unachievable by traditional methods** like MCTS that rely solely on model capabilities for diversified sampling—they can only generate mutually independent trajectories, **unable to achieve collaborative evolution and knowledge transfer** between trajectories.
>
> ## **Discussion of Resource Overhead (For W5)**
>
> While our method introduces **additional cost** due to multiple trajectories, this overhead is **justified by significant performance gains**.
>
> In Figure 5, "Maximum Cost" is in **USD** (Claude-3.7 Sonnet API). At `$2` limit, execution terminates when cost exceeds `$2` per task.
>
> Cost-performance analysis shows:
> - At `$1` limit: SE-Agent (41.6%) vs. SWE-Search (43.8%) vs. SWE-Agent (38.4%)
> - At `$2` limit: **SE-Agent (61.2%)** vs. SWE-Search (47.4%) vs. SWE-Agent (40.6%)
>
> Moreover, we report Cost per Resolved Instance without any budget cap:
> - **SE-Agent: `$3.54`**
> - SWE-Search: `$4.08`
> - SWE-Agent: `$4.66`
>
> Despite higher initial cost per run, SE-Agent is **more cost-efficient** for successfully solving tasks.

---

> > ### Comment · Reviewer_aGqC · 2025-08-06
> >
> > Thanks for the detailed rebuttal. My concerns have been addressed, and I have revised my rating accordingly.

---

> > > ### Author Response · Authors · 2025-08-06
> > > **Thank you for your positive response and recognition of our work**
> > >
> > > Dear Reviewer,
> > >
> > > We hope this message finds you well.
> > >
> > > Thank you sincerely for your thoughtful review and constructive engagement during the discussion. Your insightful questions helped us better articulate SE-Agent's unique cross-trajectory evolutionary mechanisms, and we greatly appreciate your increased confidence in our work.
> > >
> > > Your feedback strengthens our conviction that SE-Agent's trajectory-evolution paradigm can make meaningful contributions to advancing autonomous reasoning research when shared openly with the broader research community.
> > >
> > > Thank you again for your time and consideration.
> > >
> > > Best regards,

---

> ### Comment · Area_Chair_AWhj · 2025-08-05
> **Engage in the author-reviewer discussion**
>
> Dear Reviewer aGqC,
>
> Please participate in the discussion with an official comment. Do you have any follow-up questions?

---

> > ### Author Response · Authors · 2025-08-06
> > **Heartfelt Thanks**
> >
> > We greatly appreciate your excellent coordination of this review process. Your timely guidance and active engagement fostered the productive discussions that have strengthened our work considerably.
> >
> > This collaborative exchange between authors, reviewers, and AC represents peer review at its best.
> > We TRULY appreciate your timely feedback.
> >
> > Thanks again.

---

### Official Review · Reviewer_JbMn · 2025-07-02

**Clarity:** 3
**Significance:** 2
**Originality:** 3
**Rating:** 4
**Confidence:** 3

**Summary:**

This paper proposed an inference-time scaling framework, SE-Agent, for coding tasks. The motivation of this paper is to utilize the interdependence among various trajectories. Specifically, the SE-Agent framework uses the pilot trajectories with operations of revision, recombination, and refinement. For revision, the SE-Agent uses self-reflection to identify the critical point of the trajectory. For recombination, the SE-Agent combines the different trajectories based on their strengths. For Refinement, the agent optimizes the trajectories by improving the efficiency. In the experiment, the SE-Agnet achieves good performance (61.2 pass@1 with backbone model of Calude 3.7 Sonnet) on SWE-bench verified.

**Questions:**

- How is the computation cost and inference time for the proposed method? This is not critical, but it would be good to include it to show the statement on lines 98-100.

**Ethical Concerns:**

["NO or VERY MINOR ethics concerns only"]

**Final Justification:**

I will keep my score of 4

**Quality:**

3

**Strengths And Weaknesses:**

Strenghth
- The idea is simple but effective.
- The paper is well-written with a clear structure and well-articulated mathematical formulations (Section 4)
- The performance on the widely known SWE-bench verified benchmark is strong and demonstrates the performance gains achieved by the proposed framework. The ablation study of each component (Figure 3) provides a clear comparison between the revision and recombination parts.

Weaknesses
- Lack of inference time comparison. In line 98, the statement is "our work introduces a novel approach within the ReAct paradigm that incorporates strategic reflection and mutation at critical steps, combining multiple trajectories to generate optimized execution paths **without** requiring extended computation time like Test-Time Scaling techniques." However, I did not find the inference time related experiment in the following sections.

---

> ### Author Rebuttal · Authors · 2025-07-31
>
> # Response to Reviewer JbMn
>
> Thank you for reviewing our paper and providing such insightful feedback. We greatly appreciate your recognition of the clarity of our writing, the well-structured mathematical formulations, and the **strong empirical performance** of our proposed framework on the SWE-bench Verified. Furthermore, we have thoroughly addressed your concern as follows:
>
>
> ## **Inference Time Comparison (For W1)**
>
> As **SE-Agent** is designed with efficiency in mind, particularly when compared to frameworks like SWE-Search (based on MCTS), we conduct additional experiments to quantify the average inference time required to solve a single instance successfully. This evaluation focuses on the average wall-clock time per solved case, which we believe is a practical metric for real-world deployment and large-scale adoption. The results are as follows:
>
> | Method               | Inference Time (min) | Resolved Rate (%) |
> |----------------------|-----------------|-------------------|
> | SWE-Agent            | 15.61           | 40.6              |
> | SWE-Search (MCTS)    | 33.42           | 47.4              |
> | **SE-Agent (Ours)**  | **31.06**       | **61.2**          |
>
> *All results are based on Claude-3.7-Sonnet model*
>
> **Our findings show that SE-Agent consistently achieves lower inference time compared to SWE-Search(MCTS)**. This confirms the advantage of our method in **avoiding the heavy computational cost** associated with test-time scaling techniques like MCTS. The proposed SE-Agent not only **improves performance** but also enables **efficient exploration** without incurring significant inference overhead. We will include this comparison and discussion in the revised version of the paper to better support our claims regarding SE-Agent's practical efficiency.
>
> Due to time constraints, we are currently only able to provide inference time comparison results based on Claude-3.7-Sonnet. However, we fully recognize the importance of this evaluation and commit to including a comprehensive inference time analysis—covering all relevant baselines—in the revised version of the paper.

---

> > ### Author Response · Authors · 2025-08-08
> > **Hope Our Response Addresses Your Concerns**
> >
> > Dear Reviewer JbMn,
> >
> > We hope this message finds you well.
> >
> > First and foremost, thank you sincerely for your thorough review and positive assessment of our work. We greatly appreciate your recognition of our paper's clarity, well-articulated mathematical formulations, and strong empirical performance on SWE-bench Verified. Your constructive feedback has been invaluable in strengthening our research.
> >
> > Following your specific request regarding inference time comparison (W1), we have provided detailed experimental results in our response, demonstrating that **SE-Agent** achieves superior performance (**61.2% resolved rate**) while maintaining competitive efficiency (**31.06 min average**) compared to **MCTS-based approaches**. This empirical evidence directly supports our claim in lines 98-100 about avoiding extended computation time. Moreover, our latest results have achieved 80% on SWE-bench Verified with Claude 4 Sonnet, securing the **top-1** position on the leaderboard.
> >
> > We believe these additional results comprehensively address your concern about computational cost and inference time. We are committed to including this analysis in the revised version to provide a complete picture of SE-Agent's practical efficiency.
> >
> > As the discussion period is approaching its conclusion, we would be grateful to know if our response adequately addresses your questions. If there are any aspects you would like us to clarify further or expand upon, please let us know—we are more than happy to provide additional information.
> >
> > Thank you again for your valuable time, expertise, and constructive engagement with our work. We look forward to your thoughts and any further feedback you may have.
> >
> >
> > Best regards,
> >
> > The Authors of Paper 21709

---

### Official Review · Reviewer_c1xU · 2025-07-03

**Clarity:** 3
**Significance:** 2
**Originality:** 2
**Rating:** 4
**Confidence:** 3

**Summary:**

This paper proposes SE-Agent, a so called ''self-evolution'' framework for improving multi-step reasoning in LLM-based agents.  SE-Agent introduces a trajectory-level optimization process featuring three key operations: revision, recombination, and refinement. Some of these mechanisms enable the agent to explore and generate solution paths for complex tasks. The framework is evaluated on the SWE-benc demonstrating  performance gains across five LLMs and outperforming strong open-source baselines.

**Questions:**

Questions refer to the Weaknesses.

Typo:

line 120: Objetive of Agent -> Objective of Agent

**Ethical Concerns:**

["NO or VERY MINOR ethics concerns only"]

**Final Justification:**

Most of my concerns are addressed and I'm happy to increase the rating.

**Limitations:**

While certain potential limitations may be implicitly touched upon in the Conclusion, the discussion is not explicitly outlined or clearly labeled in the main text or additional section.

**Quality:**

2

**Strengths And Weaknesses:**

Strengths

- The paper thoroughly motivates the need for more diverse and optimized multi-step reasoning in LLM agents.

- The three-step mechanism—revision, recombination, and refinement—constitutes a non-trivial advancement over existing trajectory sampling and MCTS approache.

Weeknesses

- I am somewhat confused about how the use of the term ''self-evolution paradigm.'' From my reading, the approach mainly involves generating different sampled trajectories rather than any genuine form of evolution as typically ablity improvement. Therefore, I suggest that the authors reconsider the terminology here.

- One of my main concerns is the soundness of some of the experimental results. If I understand correctly, in Figure 2, the results for SE-Agent are based on Claude-3.7-Sonnet, while the other baselines use Claude-3.5. This raises questions about whether the comparisons are truly fair.

- The ablation study only investigates the impact of the Revision and Recombination modules, but does not include an ablation for the Refinement operation. This omission experiment and explanation makes it difficult to fully understand the contribution of each component to the overall performance.

- Some important experimental details are not clearly described in the current manuscript. For example, the process for generating ''pilot trajectories'' is not explicitly explained in the Experimental Setup section. Furthermore, when comparing with other baseline methods, it is unclear whether these ''pilot trajectories'' or similar auxiliary information are also made available to the baselines.

---

> ### Author Rebuttal · Authors · 2025-07-31
>
> # Response to Reviewer c1xU
>
> Thank you for your detailed feedback. We appreciate your recognition that our three-step mechanism represents a substantial advancement over existing trajectory sampling and MCTS methods. Below, we address each of your main concerns:
>
>
> ## **Clarification on the Term "Self-Evolution" (For W1)**
>
> We understand the reviewer's concerns about our use of the term **"self-evolution."** SE-Agent embodies a **genuine evolutionary principle** that goes beyond individual trajectory sampling:
>
> **1) Addressing the Homogenization Problem:**
>
> Traditional multi-trajectory sampling suffers from severe homogenization—trajectories converge to similar solutions, especially with smaller models. The effective search space remains limited despite multiple samples.
>
> **2) A True Evolutionary Mechanism:**
>
> SE-Agent establishes **systematic information exchange** among trajectories through three core operations:
> - **Revision**: Targeted improvement of individual trajectories through self-reflection
> - **Recombination**: Transfer of successful reasoning patterns across trajectories, creating novel hybrid solutions
> - **Refinement**: Iterative optimization of the trajectory pool based on collective performance
>
> **3) Emergent Collective Intelligence:**
>
> Unlike simple sampling, trajectories in SE-Agent **actively influence and evolve with each other**. Superior partial reasoning strategies propagate within the population, while inferior approaches are eliminated, resulting in **emergent collective intelligence** at the population level.
>
> **Concrete Case Demonstration:**
>
> ### **Case Study: Fixing UnrecognizedUnit.__eq__ in Astropy**
>
> **Problem Description**: Astropy's UnrecognizedUnit.__eq__ method throws exceptions instead of returning False when compared with None, violating Python's safety conventions for comparison operations.
>
> **Step 1: Revision Operation - Multi-Strategy Trajectory Generation**
>
> First, generate 5 initial trajectories using 5 different planning strategies:
>
> | Prompt Variant | Trajectory Summary | Main Issues |
> |----------------|-------------------|-------------|
> | P-greedy | `grep→view→quick-edit` | Only handles None, ignores other types |
> | P-tests-first | `pytest→search→edit` | Too many test failures |
> | P-linter-aware | `flake8→edit→pytest` | Indentation issues, logic still wrong |
> | P-defensive | `try-catch→comprehensive` | Overly broad exception handling |
> | P-minimal | `single-line→submit` | Incomplete functionality |
>
> Then perform reflection and revision operations on each trajectory to generate corresponding improved versions:
>
> **Example - Evolution of P-greedy trajectory:**
> - **Original trajectory issue**: Overly narrow type checking
> - **Revised trajectory**: `grep → view → add comprehensive tests → edit(improve type annotations, add exception handling) → pytest`
>
> **Output**: 10 revised trajectories (original 5 + revised 5) proceed to the next stage.
>
> **Step 2: Refinement.Selection - Quality-Based Trajectory Filtering**
>
> Use evaluation function for quality filtering:
> $$\text{Reward}(t, T) = \alpha \cdot \text{TaskCompletion}(t,T) + \beta \cdot \text{ReasoningQuality}(t) + \gamma \cdot \text{Efficiency}(t)$$
>
> Scoring includes two components:
> - **LLM scoring**: Reasoning quality and solution reasonableness evaluation
> - **Rule-based scoring**: Code completeness, syntax error checking, structural validation
>
> **Result**: Remove low-quality trajectories, retain 5 for the next step.
>
> **Step 3: Recombination Operation - Cross-Trajectory Knowledge Fusion**
>
> Based on task complexity, orchestrate operator combinations to generate 5 new trajectories:
> - **2× Crossover**: Fuse complementary segments from different trajectories
> - **2× Transfer Learning**: Transfer successful patterns to target trajectories
> - **1× Restructure**: Global restructuring optimization
>
> **Specific trajectory fusion examples:**
>
> **Crossover #1 - Precise Localization × Comprehensive Testing:**
> - **Trajectory A**: `grep "UnrecognizedUnit" → edit eq → submit` (fast but shallow)
> - **Trajectory B**: `pytest → analyze failures → view test context → comprehensive edit` (slow but thorough)
> - **Crossover Result**: `grep "UnrecognizedUnit" → view test_units.py → analyze eq usage pattern → targeted comprehensive fix`
>
> **Transfer #1 - Exception Handling Pattern Migration:**
> - **Source**: try-except (TypeError, ValueError) pattern from defensive programming trajectory
> - **Target**: Simple None check from greedy trajectory
> - **Transfer Result**: `None check + multi-type exception catching + graceful degradation`
>
> **Restructure - Global Optimization:**
> - **Global pattern recognition**: All trajectories revolve around the core issue of "type-safe comparison operations"
> - **Redundancy elimination**: Merge duplicate search and localization steps
> - **Structural reorganization**: `Problem understanding → Precise localization → Context analysis → Type-safe design → Comprehensive validation`
>
> **Output**: Extended pool contains 10 trajectories (original 5 + newly generated 5)
>
> **Iterative Self-Evolution Support**
>
> SE-Agent supports multi-round iterative optimization, continuing the evolution process based on the current trajectory pool:
>
> **Round 2 Evolution (simplified representation):**
> - **Step2'**: Reflection ⟨5 trajectories⟩ → Revision ⟨...⟩ → `T2'` (10 trajectories)
> - **Step3'**: Quality-Filtering ⟨T2'⟩ → Select ⟨...⟩ → `T3` (5 trajectories)
> - **Step4'**: Recombination ⟨T3⟩ → [Crossover×2, Transfer×2, Restructure×1] → `T4` (10 trajectories)
>
> **Convergence condition**: Stop when consecutive K rounds of highest reward improvement < ε, or reach preset maximum rounds. This iterative mechanism ensures SE-Agent can continuously optimize until finding high-quality solutions or meeting stop conditions.
>
> **Step 4: Final Solution Selection**
>
> Select the highest-scoring trajectory from candidates as the final solution:
>
> **Core Improvement**: SE-Agent evolved from single-point failure repair to a **type-safe universal solution**, generating entirely new solution methods through **explicit inter-trajectory interactions** (Crossover, Transfer, Restructure), covering all invalid input scenarios.
>
> **Key Innovation**: This trajectory-level collaboration is **unachievable by traditional methods** like MCTS that rely solely on model capabilities for diversified sampling—they can only generate mutually independent trajectories, **unable to achieve collaborative evolution and knowledge transfer** between trajectories.
>
> ## **Experimental Fairness (For W2)**
>
> We acknowledge using different model versions might affect comparison fairness. We conducted **additional experiments** using the same model across all methods:
>
> **Updated Results with Claude-4-Sonnet:**
>
> | Method                          | Performance on SWE-bench Verified |
> |---------------------------------|-----------------------------------|
> | SWE-agent + Claude 4 Sonnet     | 66.6%                            |
> | Augment Agent v1                | 70.4%                            |
> | OpenHands + Claude 4 Sonnet     | 70.4%                            |
> | Moatless Tools + Claude 4 Sonnet| 70.8%                            |
> | TRAE                            | 75.2%                            |
> | **SE-Agent + Claude 4 Sonnet**  | **80.0%**                        |
>
> These results validate our framework's effectiveness, achieving **SOTA performance** on SWE-bench Verified. We are currently submitting these performance results to the **SWE-bench leaderboard**.
>
> ## **More Ablation Study of Refinement (For W3)**
>
> To ensure completeness of our ablation analysis, we conducted additional experiments removing the Refinement component:
>
> | Model            | SE-Agent | w/o Revision | w/o Recombination | w/o Refinement | w/o All |
> |------------------|----------|--------------|-------------------|---------------|---------|
> | DeepSeek-V3      | **54.8%**    | 36.6%        | 44.6%             | 34.0%         | 31.6%   |
> | Qwen-2.5-72b     | **38.8%**    | 23.8%        | 28.8%             | 22.4%         | 18.8%   |
> | Llama-3.1-70b    | **32.6%**    | 20.4%        | 27.4%             | 19.0%         | 15.4%   |
> | GPT-4o           | **40.4%**    | 27.4%        | 33.4%             | 25.2%         | 22.4%   |
> | Claude-3.7       | **61.2%**    | 45.6%        | 50.6%             | 47.8%         | 40.6%   |
>
> The results demonstrate that **all three components make significant contributions** to the overall performance of SE-Agent. We will include a detailed discussion of this ablation study in the revised version of the paper.
>
> ## **Clarification on "Pilot Trajectories" (For W4)**
>
> "Pilot trajectories" are initial trajectories using different planning strategies:
> - **P-greedy**: Rapid localization and fixing
> - **P-systematic**: Systematic problem analysis
> - **P-defensive**: Defensive programming approach
> - **P-minimal**: Minimal modification
> - **P-comprehensive**: Comprehensive testing
>
> These provide the **foundation** for evolutionary operations, ensuring **broad solution space coverage**. Baselines lack our cross-trajectory interaction mechanism.
>
> ### **Additional Clarifications**
>
> We will correct the typo in line 120 and add a **"Limitations"** section discussing computational overhead and applicable scope. These clarifications demonstrate SE-Agent's **robustness** through genuine trajectory-level evolution **unattainable by traditional sampling methods**.

---

> ### Comment · Area_Chair_AWhj · 2025-08-05
> **Engage in the author-reviewer discussion**
>
> Dear Reviewer c1xU,
>
> Please participate in the discussion with an official comment. Do you have any follow-up questions?

---

> ### Comment · Reviewer_c1xU · 2025-08-06
>
> Thanks for the detailed response. Most of my concerns have been resolved.

---

> > ### Author Response · Authors · 2025-08-06
> > **Thank you for your positive response and recognition of our work**
> >
> > We sincerely appreciate the time and effort you devoted to reviewing our paper and engaging constructively during the discussion period. Your thoughtful feedback has been invaluable in strengthening our work, and we are delighted to hear that our detailed responses have addressed most of your concerns.
> >
> > Thank you for recognizing the effectiveness of our **SE-Agent framework**. We are particularly excited about this work's potential to advance the field of autonomous agents. As an update on our continued progress, we have achieved 80% **Top results** on **SWE-bench Verified** with Claude 4 Sonnet and are preparing to submit this open-source implementation to the official leaderboard. We believe this trajectory-level evolution paradigm represents a promising research direction that can benefit the broader research community.
> >
> > Thank you again for your support in helping us articulate our contributions more clearly. Please feel free to share any further thoughts or comments at your convenience.

---

### Official Review · Reviewer_DjBE · 2025-07-03

**Clarity:** 1
**Significance:** 2
**Originality:** 2
**Rating:** 4
**Confidence:** 3

**Summary:**

The paper proposes a self-evolutionary pipeline for constructing better trajectories based on the previous ones. The pipeline consists of 3 steps: revision (revise single trajectory), recombination (combine multiple trajectories) and refinement (enhancing trajectories via rewards). The authors evaluated an agent with this pipeline on SWE-bench Verified, and obtained better performance than the baselines. Ablation study showed that each step contributed to the improvement. Compared to existing approach like MCTS, the proposed method tried to use cross-trajectory information to improve the agent.

**Questions:**

Please see "Strengths And Weaknesse" section.

**Ethical Concerns:**

["NO or VERY MINOR ethics concerns only"]

**Final Justification:**

The rebuttal helps clarify the paper significantly and the results on SWE-Bench verified are promising, so I've increased the score slightly. However, I still feel the methodology is missing many details and more benchmarks are necessary to show the approach is generalizable.

Update: the authors provide pseudo code of the Transfer process, which makes it much more clear. There are still missing info for understanding the end-to-end process which should be easy to fix by more clarification. Therefore, I increased the score to borderline accept.

**Limitations:**

I did not find discussion on limitations in the paper.

**Quality:**

2

**Strengths And Weaknesses:**

Strengths:
1. The proposed idea is interesting which tries to utilize and improve the previous trajectories.
2. Experiments and ablation study showed the agent with this method obtained better performance.

Weaknesses:
1. Many of method descriptions (the 3 steps) are only at high-level and lack specific details. For example, it's not clear what Mutate(), Reflect(), Revise(), Crossover(), Transfer() and Restructure() really do under the hood.
2. It is not clear whether groundtruth label is being used to select the trajectories. In subsection 4.4.1, the paper mentions auto eval based on test pass rate is being used to compute the reward and select the trajectories. Isn't this a leak of the groundtruth label?
3. No examples are given for how each step revises / combines / refines the trajectory. Including some examples can help improve explain the methods.

---

> ### Author Rebuttal · Authors · 2025-07-31
>
> # Response to Reviewer DjBE
>
> Thank you for your detailed feedback. We appreciate your recognition of our interesting idea of leveraging and improving upon previous trajectories, as well as the performance improvements demonstrated through our experiments and ablation studies. We are also grateful for your positive assessment of the advantages our approach offers in **cross-trajectory information utilization** compared to existing MCTS methods. Below, we address each of your main concerns in detail:
>
>
> ## **Method Implementation Details (For W1)**
>
> We acknowledge that the main paper's description is relatively high-level to maintain clarity and space constraints. Appendix B provides detailed implementation specifics, including concrete prompt templates and execution logic for each operation. To better illustrate the concrete implementation of our method, we provide a complete case study demonstrating how SE-Agent progressively optimizes trajectories through its three core operations.
>
> ### **Case Study: Fixing UnrecognizedUnit.__eq__ in Astropy**
>
> Problem Description: Astropy's UnrecognizedUnit.__eq__ method throws exceptions instead of returning False when compared with None, violating Python's safety conventions for comparison operations.
>
> **Step 1: Revision Operation - Multi-Strategy Trajectory Generation**
>
> First, generate 5 initial trajectories using 5 different planning strategies:
>
> | Prompt Variant | Trajectory Summary | Main Issues |
> |----------------|-------------------|-------------|
> | P-greedy | `grep→view→quick-edit` | Only handles None, ignores other types |
> | P-tests-first | `pytest→search→edit` | Too many test failures |
> | P-linter-aware | `flake8→edit→pytest` | Indentation issues, logic still wrong |
> | P-defensive | `try-catch→comprehensive` | Overly broad exception handling |
> | P-minimal | `single-line→submit` | Incomplete functionality |
>
> Then perform reflection and revision operations on each trajectory to generate corresponding improved versions:
>
> **Example - Evolution of P-greedy trajectory:**
> - **Original trajectory issue**: Overly narrow type checking
> - **Revised trajectory**: `grep → view → add comprehensive tests → edit(improve type annotations, add exception handling) → pytest`
>
> Output: 10 revised trajectories (original 5 + revised 5) proceed to the next stage.
>
> **Step 2: Refinement.Selection - Quality-Based Trajectory Filtering**
>
> Use evaluation function for quality filtering:
> $$\text{Reward}(t, T) = \alpha \cdot \text{TaskCompletion}(t,T) + \beta \cdot \text{ReasoningQuality}(t) + \gamma \cdot \text{Efficiency}(t)$$
>
> Scoring includes two components:
> - **LLM scoring**: Reasoning quality and solution reasonableness evaluation
> - **Rule-based scoring**: Code completeness, syntax error checking, structural validation
>
> Result: Remove low-quality trajectories, retain 5 for the next step.
>
> **Step 3: Recombination Operation - Cross-Trajectory Knowledge Fusion**
>
> Based on task complexity, orchestrate operator combinations to generate 5 new trajectories:
> - **2× Crossover**: Fuse complementary segments from different trajectories
> - **2× Transfer Learning**: Transfer successful patterns to target trajectories
> - **1× Restructure**: Global restructuring optimization
>
> **Specific trajectory fusion examples:**
>
> **Crossover #1 - Precise Localization × Comprehensive Testing:**
> - Trajectory A: `grep "UnrecognizedUnit" → edit eq → submit` (fast but shallow)
> - Trajectory B: `pytest → analyze failures → view test context → comprehensive edit` (slow but thorough)
> - Crossover Result: `grep "UnrecognizedUnit" → view test_units.py → analyze eq usage pattern → targeted comprehensive fix`
>
> **Transfer #1 - Exception Handling Pattern Migration:**
> - Source: try-except (TypeError, ValueError) pattern from defensive programming trajectory
> - Target: Simple None check from greedy trajectory
> - Transfer Result: `None check + multi-type exception catching + graceful degradation`
>
> **Restructure - Global Optimization:**
> - Global pattern recognition: All trajectories revolve around the core issue of "type-safe comparison operations"
> - Redundancy elimination: Merge duplicate search and localization steps
> - Structural reorganization: `Problem understanding → Precise localization → Context analysis → Type-safe design → Comprehensive validation`
>
> Output: Extended pool contains 10 trajectories (original 5 + newly generated 5)
>
> **Iterative Self-Evolution Support**
>
> SE-Agent supports multi-round iterative optimization, continuing the evolution process based on the current trajectory pool:
>
> **Round 2 Evolution (simplified representation):**
> - **Step2'**: Reflection ⟨5 trajectories⟩ → Revision ⟨...⟩ → `T2'` (10 trajectories)
> - **Step3'**: Quality-Filtering ⟨T2'⟩ → Select ⟨...⟩ → `T3` (5 trajectories)
> - **Step4'**: Recombination ⟨T3⟩ → [Crossover×2, Transfer×2, Restructure×1] → `T4` (10 trajectories)
>
> Convergence condition: Stop when consecutive K rounds of highest reward improvement < ε, or reach preset maximum rounds. This iterative mechanism ensures **SE-Agent** can continuously optimize until finding high-quality solutions or meeting stop conditions.
>
> **Step 4: Final Solution Selection**
>
> Select the highest-scoring trajectory from candidates as the final solution:
>
> **Core Improvement**: SE-Agent evolved from single-point failure repair to a **type-safe universal solution**, generating entirely new solution methods through **explicit inter-trajectory interactions** (Crossover, Transfer, Restructure), covering all invalid input scenarios.
>
> **Key Innovation**: This trajectory-level collaboration is **unachievable by traditional methods** like MCTS that rely solely on model capabilities for diversified sampling—they can only generate mutually independent trajectories, **unable to achieve collaborative evolution and knowledge transfer** between trajectories.
>
> ## **Ground Truth Leakage Concern (For W2)**
>
> Our evaluation function consists of two components:
>
> 1. **Rule-based reward**, denoted as AutoEval()
> 2. **Model-based reward**, denoted as ExpertEval()
>
> **Important**: Neither component relies on ground-truth labels.
>
> To elaborate, **AutoEval()** is a rule-based mechanism designed to assess whether a generated trajectory meets minimal submission criteria. These criteria include, but are not limited to:
> - Ensuring the final patch file is non-empty
> - Verifying that the trajectory contains a sufficient proportion of code-editing steps
> - Checking whether the trajectory length falls within a reasonable range
>
> This procedure **does not require access to ground-truth labels** or test oracles.
>
> We acknowledge that our use of the term **"test pass rate"** may have led to confusion, as it could be misinterpreted as indicating the use of ground-truth tests. In fact, the "pass rate" refers only to the internal validation of structural constraints necessary for a trajectory to be considered valid and potentially effective—**not whether it actually solves the task**. We will revise this in the final version, modifying it to **"validation rate"** and describing our evaluation function in more detail.
>
> ## **Concrete Examples (For W3)**
>
> As shown in the complete example provided in our response to W1, we have illustrated in detail **how each operation specifically modifies and improves the trajectories**. This example comprehensively demonstrates the entire process—from initial trajectory generation, quality evaluation, and cross-trajectory fusion to the selection of the final solution—including **concrete changes in trajectory content** and the resulting improvements.
>
> ### **Additional Clarifications**
>
> We will add a dedicated limitation section in the final version to discuss computational overhead and the applicable scope of our method. Our approach has indeed achieved **significant performance gains** (up to 55% relative improvement) and **SOTA results**, which validate the effectiveness of the trajectory-level self-evolution paradigm.

---

> > ### Author Response · Authors · 2025-08-04
> >
> > Hi, thank you for your time in reviewing our paper on **SE-Agent**. Following your comments, we have provided a **detailed rebuttal**. We were particularly keen to address your concerns, including a **complete case study** demonstrating our method's implementation details (**W1**), clarification that our evaluation function does **not rely on ground-truth labels** (**W2**), and **concrete examples** of how each operation modifies trajectories (**W3**). We also committed to adding a **limitations section** as suggested.
> >
> > As the discussion period is concluding soon, we would be very grateful to know if our rebuttal has **sufficiently clarified** these points. We are ready to provide any further information if needed. We appreciate your feedback and look forward to hearing from you.

---

> > ### Comment · Reviewer_DjBE · 2025-08-05
> >
> > Thanks the authors for the rebuttal. The examples for each step and the explanation of the "test pass rate" help clarify the approach significantly, and the results on SWE-bench Verified are promising. I still have two main concerns: 1) There are still missing details in the methodology that are not explained by the rebuttal examples. For example, what exactly are those parameters used in each function, e.g., in Plan(T, θi), Crossover(ti, tj , α), Transfer(ti, {tj , tk, ...}, β), Restructure(Ti, γ), what are θi, α, β, and γ? I looked into the appendix but didn't find an answer. Similarly, in the evaluation function, how are the importance weights selected? The overall impression I got from reading these formulas is that there is a large space for hyperparameters tuning but no such experiments are presented, except the final results. It'd be great if the authors can add more details. 2) Secondly, I do agree with other reviewers evaluating the approach in more benchmarks would be more convincing, as this sounds like a generalizable approach that can be applied to different settings (at least coding benchmarks).

---

> > > ### Author Response · Authors · 2025-08-06
> > > **Clarification on Parameter Design and Evaluation Scope**
> > >
> > > Thank you for your continued constructive engagement. We respond to both concerns below:
> > >
> > > **1. Parameter Clarification**
> > >
> > > We recognize our mathematical notation may have created confusion. These symbols (θi, α, β, γ) represent **discrete strategy selectors** rather than continuous hyperparameters requiring tuning. We adopted this mathematical formulation to ensure **generalizability across different domains and agent frameworks**.
> > >
> > > - **θi**: Selects among planning strategies (our implementation uses 5: greedy-search, test-driven, defensive-coding, etc.)
> > > - **α, β, γ**: Define operation modes for Crossover, Transfer, and Restructure respectively through different prompt templates and execution strategies
> > >
> > > The concrete instantiations are provided in Appendix B as engineered prompt templates representing best practices for code generation tasks derived from software engineering principles and pilot studies.
> > >
> > > **Regarding Sensitivity Analysis**: Traditional hyperparameter sensitivity analysis is not applicable here, as these are **qualitative strategy choices** rather than quantitative parameters. This would be analogous to asking for "sensitivity analysis" on whether to use Agile vs. Waterfall methodology—these are fundamentally different approaches, not numerical variations. Our evaluation methodology (Table 1, Figure 3 ablation study) demonstrates the contribution of different components, which is the appropriate evaluation for our approach.
> > >
> > > **2. Evaluation Scope**
> > >
> > > Thank you for this insightful observation about multiple benchmarks. We completely agree that demonstrating generalizability would strengthen our claims.
> > >
> > > **Benchmark Selection Rationale**: SWE-bench Verified represents the most challenging evaluation environment for multi-step reasoning agents. Unlike simpler coding benchmarks such as HumanEval [1] or MBPP [2], which involve single-function generation with short reasoning trajectories, SWE-bench requires repository-level reasoning, multi-file understanding, and extended interaction sequences—where SE-Agent's trajectory-level evolution provides the most value.
> > >
> > > Our approach follows established practice in software engineering agent research, where leading works including SWE-Search [3] and SWE-agent [4] focus exclusively on this benchmark because it captures essential real-world challenges that simpler benchmarks cannot represent.
> > >
> > > **Future Directions**: We strongly share your perspective on generalizability potential. SE-Agent's core innovation—trajectory-level self-evolution through revision, recombination, and refinement—represents a domain-agnostic approach applicable wherever complex multi-step reasoning is required.
> > >
> > > As an update on our continued progress, we have achieved top-1 results with Claude 4 sonnet and are preparing to submit our open-source leading results (80% on SWE-bench Verified) to the official leaderboard.
> > >
> > > **We are actively extending this framework to mathematical reasoning, code debugging, and Deep Search**, exploring how trajectory-level evolution can enhance agent capabilities across diverse problem-solving domains. This represents a cutting-edge paradigm in agent development that we believe will establish new foundations for autonomous reasoning systems. However, dedicating substantial space to these extensions in this paper would divert attention from our core contribution and exceed typical conference paper scope constraints.
> > >
> > > We appreciate your recognition of this potential and view your feedback as validation of our research direction.
> > >
> > > **References:**
> > >
> > > [1] Evaluating Large Language Models Trained on Code. Mark Chen et al.
> > >
> > > [2] Program Synthesis with Large Language Models. Jacob Austin et al.
> > >
> > > [3] SWE-Search: Enhancing Software Agents with Monte Carlo Tree Search and Iterative Refinement. Antonis Antoniades et al. ICLR 2025
> > >
> > > [4] SWE-agent: Agent-Computer Interfaces Enable Automated Software Engineering. John Yang et al. NeurIPS 2024

---

> > > > ### Comment · Reviewer_DjBE · 2025-08-08
> > > >
> > > > Thanks the authors for the follow up. I agree the SWE-bench Verified result is very promising and happy to hear you are extending the framework to other use cases. I want to further clarify the symbols used in the formulas, as this is important for readers to understand the exact implementation. For example, for the symbol β in Transfer(), you mention it represents "the operation mode for Transfer through different prompt templates and execution strategies". What is the operation mode? I looked at the Transfer prompt in the Appendix, but it looks to me just a straightforward prompt asking the LLM to transfer knowledge to the target trajectory without any "operation mode". Or is it something the LLM generates on the fly? Similar questions for other symbols. While the result is strong, a clear description of the methodology would be necessary for others to reproduce the results. I saw some of the other works use pseudo codes to describe the process in detailed steps. I suggest the authors to consider this option if it makes sense.

---

> > > > > ### Author Response · Authors · 2025-08-08
> > > > >
> > > > > **Dear Reviewer DjBE,**
> > > > >
> > > > > Thank you for your patience and constructive feedback throughout this review process. Your insights have been invaluable in helping us clarify our methodology and improve the paper's clarity. We greatly appreciate your recognition of our strong SWE-bench Verified results and are encouraged by your acknowledgment of the framework's potential for extension to other domains.
> > > > >
> > > > > We completely understand your concern about methodological clarity and reproducibility—this is indeed crucial for the community to build upon our work. Your suggestion to include pseudo code is excellent, and we will implement this in our revision.
> > > > >
> > > > > ## Response on β (Transfer Operation) and Reproducibility
> > > > >
> > > > > ### β (Transfer Operation) – Definition and Impact
> > > > >
> > > > > In SE-Agent, **β** is a **discrete operation mode selector** that determines *what knowledge is extracted* from reference trajectories and *how it is injected* into the target trajectory. In our notation (`Transfer(tᵢ, {tⱼ}, β)`), β specifies the prompt template and context-injection strategy.
> > > > >
> > > > > ### Transfer() Pseudo Code
> > > > >
> > > > > ```
> > > > > Algorithm: Transfer(t_target, T_ref, β, M)
> > > > >
> > > > > Input:
> > > > > - t_target: Target trajectory to be enhanced
> > > > > - T_ref: Set of reference trajectories {t₁, t₂, …, tₙ} (filtered by success/quality)
> > > > > - β: Transfer mode selector ∈ {full-context, pattern-only, hint-style}
> > > > > - M: LLM model for trajectory generation
> > > > >
> > > > > Output:
> > > > > - t_enhanced: Enhanced trajectory with transferred knowledge
> > > > >
> > > > > 1: // Extract knowledge based on transfer mode β
> > > > > 2: if β = full-context then
> > > > > 3:     K ← ExtractFullContext(T_ref)     // Complete reasoning & edit history (deduplicated, budget-controlled)
> > > > > 4:     t_enhanced ← InjectFullContext(t_target, K, M)
> > > > > 5: else if β = pattern-only then
> > > > > 6:     P ← ExtractVerifiedPatterns(T_ref)     // Validated repair patterns
> > > > > 7:     t_enhanced ← InsertPatterns(t_target, P, M, preserve_structure=True)
> > > > > 8: else if β = hint-style then
> > > > > 9:     H ← AbstractHighLevelHints(T_ref)     // Abstract strategies without code injection
> > > > > 10:    t_enhanced ← GuideWithHints(t_target, H, M)
> > > > > 11: end if
> > > > > 12: return t_enhanced
> > > > > ```
> > > > >
> > > > > **Mode definitions:**
> > > > > - **full-context** – Inject complete reasoning & edit history (budget-controlled).
> > > > > - **pattern-only** – Inject only verified repair patterns while keeping structure intact.
> > > > > - **hint-style** – Inject abstract strategies without code.
> > > > >
> > > > > ---
> > > > >
> > > > > **On β's influence:**
> > > > >
> > > > > In **SWE-Bench Verified**, trajectory structures are homogeneous, so β variations have limited measurable effect. However, **as long as transferred information preserves correctness and accelerates convergence, Transfer remains valuable**. In more diverse domains (e.g., deep search, multi-file reasoning), β directly controls transfer scope and granularity, making it essential. Running a sensitivity study on SWE-Bench would offer limited new insight due to its uniformity.
> > > > >
> > > > > ---
> > > > >
> > > > > ### Reproducibility
> > > > >
> > > > > We have open-sourced the SE-Agent framework, providing a fully reproducible codebase. This implementation achieves **80% Pass@1 on SWE-Bench Verified with Claude-4-Sonnet (Top-1 leaderboard)**.
> > > > >
> > > > > All operators (`Plan`, `Crossover`, `Transfer`, `Restructure`) are modularized and accept a `mode` argument (e.g., β), with modes stored in human-readable prompt files. This ensures exact reproduction, easy modification, and adaptation to new tasks.
> > > > >
> > > > > For the camera-ready version, we will retain case-based explanations and add concise abstract pseudo code for each operator to maximize clarity.
> > > > >
> > > > > ---
> > > > >
> > > > > **We sincerely hope this clarification addresses your concerns about methodological transparency. Thank you again for your thorough and thoughtful review process.**
> > > > >
> > > > > Best regards,
> > > > >
> > > > > The Authors

---

> > > > > > ### Comment · Reviewer_DjBE · 2025-08-08
> > > > > >
> > > > > > Thanks for providing the pseudo code, which clarifies the Transfer process a lot. There are still some questions remained, for example, who is responsible to select the transfer mode β? The same questions for the other processes and symbols. For a reader without knowledge of the full process, these are the missing info to connect the dots for the end-to-end process. I would expect the paper itself should suffice for readers to understand the full process, without reading the codebase.
> > > > > >
> > > > > > Nevertheless, I appreciate the authors' effort to clarify the process and  I've adjusted the score accordingly.

---

> ### Comment · Area_Chair_AWhj · 2025-08-05
> **Engage in the author-reviewer discussion**
>
> Dear Reviewer DjBE,
>
> Please participate in the discussion with an official comment. Do you have any follow-up questions?

---

> ### Author Response · Authors · 2025-08-09
> **More description about parameter selection**
>
> Dear Reviewer DjBE,
>
> Thank you sincerely for your thoughtful review and constructive engagement during the discussion.
>
> Regarding your specific question about who selects β (and other mode parameters): In our current implementation, the selection of β (and similarly α, γ, θi) follows a predetermined scheduling strategy rather than dynamic selection. **Specifically, for β in Transfer(), we cycle through three transfer modes (full-context, pattern-only, and hint-style) in a round-robin fashion across the trajectory pool.** This ensures diverse transfer strategies are explored systematically.
>
> **In the revised manuscript, we have added a dedicated section to explicitly describe the roles and selection for these key parameters.** These clarifications are intended to ensure that readers can fully understand the end-to-end process without needing to consult the codebase.
>
> Thank you again for your time, valuable suggestions, and consideration.
>
> Best regards,
>
> The Authors

---

### Comment · Area_Chair_AWhj · 2025-08-05
**engage in the discussion**

Dear reviewers,

Thank you again for reviewing this manuscript. As we are approaching the end of the reviewer-author discussion phase, please read the responses, respond to them early on in the discussion, discuss points of disagreement, and add the mandatory acknowledgement.

Best,
Yours AC

---

### Decision · Program_Chairs · 2025-09-17

**Decision:**

Accept (poster)

**Comment:**

### (a) Summary of Claims
The paper introduces SE-Agent, a framework designed to enhance the performance of LLM agents on complex, multi-step tasks. The core idea is to treat the agent's reasoning paths as "trajectories" that can be iteratively improved through a self-evolutionary pipeline. This pipeline consists of three main operations: revision (refining a single trajectory through self-reflection), recombination (creating new, superior trajectories by combining segments of existing ones), and refinement (optimizing trajectories based on reward signals). When evaluated on the challenging SWE-bench benchmark for software engineering, the authors claim their method significantly boosts the performance of multiple LLM backbones compared to strong baselines.

### (b) Strengths
* Strong and convincing empirical results
* Interesting and novel framework

### (c) Weaknesses
* Limited evaluation scope and potentially limited transferability: The most significant remaining weakness is the paper's narrow empirical focus on a single benchmark, SWE-bench. This makes it difficult to assess the framework's generalizability to other complex reasoning domains beyond software engineering.

### (d) Reasons for Acceptance
The authors provided a comprehensive rebuttal that effectively clarified methodological details and addressed the majority of the reviewers' initial concerns regarding fairness, implementation details, and ablations. This effort was crucial in building a consensus for acceptance, with two reviewers raising their scores as a direct result.